



# Mitigating Hail Overforecasting in the 2-Moment Milbrandt-Yau Microphysics Scheme (v2.25.2_beta_04) in WRF (v4.5.1) by Incorporating the Graupel Spongy Wet Growth Process (MY2_GSWG v1.0)

Shaofeng Hua[1,2], Gang Chen[3*], Baojun Chen[1,2], Mingshan Li[4], and Xin Xu[5,6]

[1]China Meteorological Adminstration Key Laboratory of Cloud-Precipitation Physics and Weather Modification, Beijing, 100081, China
[2]China Meteorological Adminstration Weather Modification Centre, Beijing, 100081, China
[3]Nanjing Joint Institute for Atmospheric Sciences, Nanjing, 210009, China.
[4]Jingmen Meteorological Service, Jingmen, 448000, China.
[5]Key Laboratory of Mesoscale Severe Weather/Ministry of Education, and School of Atmospheric Sciences, Nanjing University, Nanjing, 210023, China.
[6]CMA Radar Meteorology Key Laboratory, Nanjing, 210023, China.

*Correspondence to*: Gang Chen (chengang@cma.gov.cn)

**Abstract.** Hail forecasting using numerical models remains a challenge due to the uncertainties and deficiencies in microphysics schemes. In this study, we assessed the hail simulation performance of the 2-moment Milbrandt-Yau (MY2) microphysics scheme within the Weather Research and Forecast (WRF) model by simulating three heavy rainfall events in
Meiyu systems in which hail was rarely observed, rather than focusing on hail cases as done in previous researches. Simulation results showed that MY2 scheme produced noticeable hail in these rainstorms. Further analysis revealed that the overprediction of hail was caused by the imperfect graupel-to-hail conversion parameterization method adopted in the MY2 scheme. By incorporating the graupel spongy wet growth process, the modified scheme significantly mitigated the hail overforecasting. Moreover, the modified MY2 scheme kept the ability to simulate hail in real hail cases, demonstrating its
ability to differentiate between heavy rainfall and hail events—a distinction the original scheme lacked. By comparing simulations of both rainstorms and hailstorms, it is concluded that the upward transport of large raindrops near the 0 °C level is critical for the graupel-to-hail conversion.

## 1 Introduction

Hailstorms are high-impact weather events with significant damage potential, amounting to billions of U.S. dollars in losses
annually around the world (Zhang et al., 2008; Changnon, 2009; Brown et al., 2015; Kunz et al., 2018). However, accurately predicting hail remains a challenge in the current numerical weather prediction models, due to the need for precise



representations of environmental conditions, the dynamic structure of hailstorms, and the associated microphysical processes (Wellmann et al., 2020).

Traditional hail forecasting methods primarily rely on environmental parameters derived from soundings to predict the
likelihood of severe thunderstorms, rather than explicitly forecasting the conditions favorable for hail formation and development (Allen et al., 2011). Moreover, predicting hail size using sounding-derived parameters is particularly challenging, with limited success in previous studies (Smith et al., 2012; Tuovinen et al., 2015). Hailstones are influenced not only by large-scale environmental conditions but also by the intensity of the parent hailstorm, the structure of the wind field, and the microphysical processes involved in hail development. These processes typically occur on small spatial and
temporal scales, while sounding data represent large-scale, time-averaged meteorological conditions. As a result, these methods often fail to accurately capture the complex mechanisms underlying hail formation and development (Allen et al., 2019).

In recent years, the application of numerical models in hail forecasting has gained increasing attention. The 1D hail growth scheme, called the HAILCAST scheme (Adams-Selin and Ziegler 2016), has shown reasonable accuracy in predicting hail
size (Gagne et al. 2017; Adams-Selin et al. 2019, 2023). However, this scheme does not directly simulate the formation of hail or the growth of hail populations. Instead, it places five hail embryos at predefined vertical levels in suitable locations once some specific dynamic conditions are met, and then assesses how large these hailstones will grow through a series of microphysical processes. Since this scheme does not calculate microphysical processes associated with hail populations, it does not provide feedback on the dynamic, thermal, and other hydrometeor fields. Furthermore, accurate simulation of hail
diameter in different regions and cases depends on parameter tuning within the HAILCAST scheme (Adams-Selin et al. 2019).

Using microphysical schemes that can simulate hail populations (referred to as the Hail Microphysical Scheme in this study, or HMS) is a more reasonable and comprehensive approach to simulating hail cases, especially in terms of the mutual feedback between thermodynamics and cloud microphysics. Previous studies have demonstrated that incorporating hail and
associated microphysical processes could improve the simulation of storm structure, cold pool intensity, and radar echo morphology (Bryan and Morrison, 2012; Tao et al., 2016). Employing HMSs allows for forecasting hail size, investigating the mechanisms of hail formation, and exploring the impact of environmental conditions on hail formation and evaluation (e.g., Milbrandt and Yau, 2006; Dennis and Kumjian, 2017; Li et al., 2017; Yin et al., 2021; Wu et al., 2022; Dong et al., 2024).

Before these microphysics schemes can be reliably used, their hail simulation performance must be evaluated against observations. Previous researches often evaluate the accuracy of hail simulation by simulating specific hail cases and comparing model-diagnosed hail size and hail coverage with those derived from operational radar data through the maximum estimated size of hail (MESH) algorithm (Witt et al., 1998) or hydrometeor classification algorithms (HCAs; Park



et al. 2009; Ortega et al. 2016). For example, Luo et al. (2017, 2018) simulated two hail events in East China using different
configurations of the Milbrandt scheme (Milbrandt and Yau, 2005a, 2005b) and found that the three-moment Milbrandt
scheme yielded surface hail diameters and hail coverage areas that more closely matched observations. However, they also
noted that the Milbrandt scheme inaccurately simulated larger hail diameters below the freezing level. Labriola et al. (2019a)
demonstrated that this overestimation of hail diameter is due to the continued growth of hail by collecting cloud droplets and
raindrops below the melting layer. Additionally, Labriola et al. (2017, 2019a, 2019b) compared observed and simulated hail
diameters across multiple hail events and identified various issues with multiple HMSs. For example, the NSSL scheme
(Mansell et al., 2010) failed to simulate large hail, the Milbrandt scheme showed excessive melting at lower levels, and the
P3 scheme (Morrison and Milbrandt, 2015) simulated high-density ice particles only near the surface, while low-density ice
particles dominated above the melting layer.

However, there are some issues with quantitatively evaluating the model's ability to simulate hail size based on the
observations. First, methods like MESH and HCA just estimate hail diameter based on radar data, which are not direct
measurements of hail size. Second, the HMS does not directly forecast hail diameter; instead, it diagnoses the maximum hail
diameter using a method that derives it from the predicted mass and number mixing ratios of hail particle populations. The
parameter in this diagnostic method is artificially set, introducing uncertainties in the diagnosed hail diameter. Furthermore,
numerous microphysical processes can influence the simulated hail diameter, making it challenging to pinpoint which
specific process is responsible when the simulation produces an inaccurate hail diameter.

A more effective and fundamental approach to evaluate HMSs is to qualitatively assess the presence or absence of hail.
Binary classification of hail presence based on HCA is more reliable than estimated hail size. Moreover, focusing on the
presence or absence of hail when evaluating microphysics schemes shifts the emphasis to processes associated with hail
formation, making it easier to identify the sources of simulation biases. We believe that a robust HMS should at least reliably
distinguish between the presence and absence of hail, accurately producing hail in cases where it forms and not producing it
in cases where it does not.

In this study, we evaluate the capability of a selected HMS (MY2 scheme) to simulate the presence or absence of hail using
data from heavy rainfall events within the Meiyu system in East China. Previous researches have shown that in typical
Meiyu systems characterized by high total precipitable water and uniformly moist conditions, heavy rainfall is more likely to
occur, while hail is rarely observed (Chen et al., 2019; Wang et al., 2019; Yao et al., 2020). These rainstorms with no hail
provide excellent examples for examining the ability of HMS to simulate hail. The description of the model configuration,
cases, and data used in this study is provided in the next section. Section 3 details the model evaluation results, modifications
to the selected hail microphysics scheme, and reevaluation of the modified scheme using data from both heavy rainfall and
hail events. Section 3 also includes an interpretation of why hail forms in hail events but does not form in heavy rainfall
events. Finally, conclusions and discussions regarding the modified HMS are presented in Section 4.



## 2 Model and Methods

### 2.1 Model configuration

This study employs the Advanced Weather Research and Forecasting (WRF; Skamarock et al., 2021) model version 4.5.1 to simulate three heavy rainfall events and three hail events occurred in East China. The 0.25° × 0.25° 1-hour-interval ERA5 reanalysis data is used as the initial and boundary conditions for each case. The digital filter initialization method is adopted to remove initial model imbalance. Simulations are conducted with three 2-way nested domains, with the horizontal grid spacings being 9 km, 3 km and 1 km, respectively. The time step for the innermost domain is 2 s. All the simulation analyses are based on the results provided by the innermost 1-km-spacing domain. Details about the model configuration for each case are shown in Table 1. The Noah Multiphysics land surface model (Niu et al., 2011), Rapid Tadiative Transfer Model for GCMs (RRTMG) radiation scheme (Pincus et al., 2003) are used for all simulations. The MYJ planetary boundary layer scheme (Janjic, 1994) is used when simulating heavy rainfall events, whereas the ACM2 scheme (Pleim, 2007) was chosen for hail simulations due to its more accurate representation of the timing and location of convective initiation compared to observations.

Table 1. Model configurations.

| | Rainstorm1 | Rainstorm2 | Rainstorm3 | Hailstorm1 | Hailstorm2 | Hailstorm3 |
|---|---|---|---|---|---|---|
| Driving data | ERA5 reanalysis | | | | | |
| Simulation time (UTC) | 0600 23 – 0600 24 June 2022 | 0600 17 – 0600 18 June 2023 | 0600 06 – 0600 07 July 2023 | 0000 – 1200 30 April 2021 | 0000 – 1200 24 June 2022 | 0000 – 1200 25 March 2024 |
| Time step | 2s in the innermost domain | | | | | |
| Horizontal resolution | 9, 3, 1 km spacing | | | | | |
| Vertical coordinate | 75 terrain-following ETA levels | | | | | |
| Land surface model | Noah Mp | | | | | |
| Radiation | RRTMG shortwave and longwave radiation model | | | | | |
| PBL scheme | MYJ | | | ACM2 | | |
| Microphysics | Original and Modified MY2 schemes | | | | | |


This study selected the MY2 scheme for evaluating hail simulations due to its recognition as a classic bulk microphysics scheme known for its ability to forecast hail. This scheme has significantly influenced the development of other microphysics schemes. For example, the graupel-to-hail conversion parameterization from the MY2 scheme, as mentioned in this study, has been adopted and further refined by the NSSL, NTU (Tsai and Chen, 2020), and SBM (Khain et al., 2010a) schemes. Following Labriola et al. (2019a), the accretion of hail by collecting liquid water beneath the melting layer is prohibited in this study, to avoid the unreasonable increase of hail size within this layer.



In addition to surface hail reports, the observed hail occurrence facts are also obtained from the hydrometeor classification algorithm (HCA) applying to S-band polarimetric radar. The HCA in this study comes from Dolan et al. (2013), which is a fuzzy-logic algorithm based on the construction of membership functions for each hydrometeor type using multiple polarimetric radar variables, including radar reflectivity ($Z_H$), differential reflectivity ($Z_{DR}$), specific differential phase ($K_{DP}$), and copolar correlation coefficient ($\rho_{hv}$). The dominant hydrometeor type in a given radar sample volume is identified (Vivekanandan et al., 1999). Although ten types of hydrometeors are identified by the HCA, whether the identified type is hail is the primary concern of this paper. Hail and graupel are primarily distinguished by the combined distribution ranges of multiple polarimetric radar variables. For example, the radar reflectivity factor ($Z_H$) for graupel is typically in the range of 34.1-54.5 dBZ, while for hail, it is between 48-76.6 dBZ. The differential reflectivity ($Z_{DR}$) for graupel is usually between 0.4-2.8 dB, whereas for hail, it is between -0.4 to 0.7 dB.

**2.2 Modification of the MY2 scheme**

According to the MY2 scheme, graupel particles that have just converted to hail through wet growth are envisioned as ice balls with a loose ice core with a bulk density of 400 kg m$^{-3}$, covered by a thin film of unfrozen water. However, real hailstone images show that even small hailstones, which primarily undergo dry growth and lack transparent layers, have a hard and high-density ice core. Graupel particles have internal voids, resulting in a bulk density lower than that of pure ice. During wet growth, unfrozen supercooled water first infiltrates these voids and freezes, increasing the bulk density of the graupel. Then, the remaining unfrozen water forms a film on the surface. This process is known as spongy wet growth (Pruppacher and Klett, 1997; Khain and Pinsky, 2018). Most microphysics schemes do not account for this process. Filling the internal voids requires more supercooled water, which can prevent graupel particles from converting to hail too easily near the 0 °C level, which will be shown in the following analysis.

Furthermore, larger particles collect more supercooled water upon collision, making it more likely to undergo wet growth. Therefore, when the amount of supercooled water is insufficient to activate wet growth for all the graupel particles with different sizes, only those larger than a certain size will undergo wet growth, while smaller particles continue to undergo dry growth. In the modified MY2 scheme, we specify that graupel particles will only convert to hail if, the total amount of supercooled cloud droplets and raindrops collected by particles larger than a critical size, exceeds the amount required for their wet growth and for filling their internal voids to achieve a density of 900 kg m$^{-3}$ within one integration timestep. We use the incomplete gamma function to describe the integration over the graupel spectrum from a certain particle size D* to infinity. The total amount of supercooled cloud droplets collected by graupel particles larger than D* per second is:

$$Q_{cg} = \frac{\pi}{4} \int_{D^*}^{\infty} \int_0^{\infty} |V_g(D_g) - V_c(D_c)| (D_g + D_c)^2 m_c(D_c) E_{cg} N_g(D_g) N_c(D_c) dD_c dD_g \, , \tag{1}$$





where $D_g$ ($D_c$), $V_g$ ($V_c = 0$), $N_g$ ($N_c$) are the diameter, terminal fall velocity, size distribution of graupel (cloud), respectively. $V_g = (\frac{\rho_0}{\rho_{air}})^{\frac{1}{2}}a_g D_g^{b_g}$, where $\rho_{air}$ is air density and $\rho_0 = 1.225$ kg m⁻³. $V_c = 0$. $m_c = \frac{\pi}{6}\rho_w D_c^3$ is the mass of a cloud droplet with diameter $D_c$, where $\rho_w$ is water density. $E_{cg}$ is the collecting efficiency between graupel and cloud. In the Milbrandt scheme, cloud size distribution is parameterized with $N_c(D) = N_{0c}D^{\alpha_c}e^{-\lambda_c D}$, where $N_{0c} = N_{Tc}\frac{1}{\Gamma(1 + \alpha_c)}\lambda_c^{1+\alpha_c}$, where $N_{Tc}$ is the total number concentration and $\alpha_c = 1$. By substituting these equations and integrating them over the cloud and graupel spectrum, we can obtain:

$$Q_{cg} = (\frac{\rho_0}{\rho_{air}})^{\frac{1}{2}}a_g\rho_w\frac{\pi^2}{24}E_{cg}\frac{N_{Tc}N_{0g}}{\Gamma(1 + \alpha_c)}\left[\frac{\Gamma(3 + b_g,\lambda_g D^*)\Gamma(1 + \alpha_c + \frac{3}{v_c})}{\lambda_c^3} + \frac{2\Gamma(2 + b_g,\lambda_g D^*)\Gamma(1 + \alpha_c + \frac{4}{v_c})}{\lambda_c^4} + \frac{\Gamma(1 + b_g,\lambda_g D^*)\Gamma(1 + \alpha_c + \frac{5}{v_c})}{\lambda_c^5}\right].$$

(2)

Similarly, the total amount of supercooled raindrop collected by graupel particles larger than D* per second is:

$$Q_{rg} = \frac{\pi}{4}\int_{D^*}^{\infty}\int_0^{\infty}|V_g(D_g) - V_r(D_r)|(D_g + D_r)^2m_r(D_r)E_{rg}N_g(D_g)N_r(D_r)dD_r dD_g,$$

(3)

according to Murakami (1990) $|v_g(D_g) - v_r(D_r)| \approx \sqrt{(V_{Qg} - V_{Qr})^2 + 0.04V_{Qg}V_{Qr}}$, where $V_{Qg}$ and $V_{Qr}$ are the mass-weighted mean fall velocity of graupel and rain, respectively. Rain size distribution is also parameterized as: $N_r(D) = N_{0r}D^{\alpha_r}e^{-\lambda_r D}$, where $\alpha_r = 0$. Thus, we can get:

$$Q_{rg} = (\frac{\rho_0}{\rho_{air}})^{\frac{1}{2}}a_g\rho_w\frac{\pi^2}{24}E_{rg}N_{Tr}N_{0g}e^{-\lambda_g D^*}\left[\frac{\Gamma(4)}{\lambda_r^3}(\frac{D^{*2}}{\lambda_g} + \frac{2D^*}{\lambda_g^2} + \frac{2}{\lambda_g^3}) + \frac{2\Gamma(5)}{\lambda_r^4}(\frac{D^*}{\lambda_g} + \frac{1}{\lambda_g^2}) + \frac{\Gamma(6)}{\lambda_r^5}\frac{1}{\lambda_g}\right].$$

(4)

Based on Musil (1970), Lin et al (1983) and Milbrandt and Yau (2005a), the critical amount of supercooled water required for graupel particles larger than D* to undergo wet growth is:

$$Q_{gwet} = \int_{D^*}^{\infty}2\pi D_g Vent\frac{\rho_{air}L_v D_v\Delta\rho - K_a T_c}{L_f + C_w T_c}dD_g,$$

(5)

where $L_v$ is the latent heat of vaporization, $D_v$ is the molecular diffusion coefficient of water, $K_a$ the thermal conductivity of air, $T_c$ the Celsius temperature, $C_w$ the specific heat of water. Vent is bulk ventilation coefficient for graupel, which is parameterized as $Vent = 0.78 + 0.308Sc^{\frac{1}{3}}Re^{\frac{1}{3}}$, where Sc is the Schmidt number, Re the Reynolds number. $Re = \frac{D_g V_g \rho_{air}}{\mu}$, where μ is the dynamic viscosity of air. By substituting these equations and integrating them over the graupel spectrum from particles larger than D*, we can obtain:



$$Q_{gwet} = 2\pi \frac{\rho_{air}L_v D_v \Delta\rho - K_a T_c}{L_f + C_w T_c} N_{0g}[\frac{0.78e^{-\lambda_g D^*}}{\lambda_g^2} + 0.308Sc^{\frac{1}{3}}\left(\frac{\rho_{air}}{\mu}\left(\frac{\rho_0}{\rho_{air}}\right)^{\frac{1}{2}} a_g\right)^{\frac{1}{2}} \frac{\Gamma\left(2.5 + \frac{b_g}{2}, \lambda_g D^*\right)}{\lambda_g^{2.5 + \frac{b_g}{2}}}].$$

(6)

The supercooled water mass required to fill the internal voids of graupels larger than D* per timestep is:

$$Q_{fill} = \int_{D^*}^{\infty} \frac{\pi}{6}(\rho_h - \rho_g)D_g^3 N_g(D_g)dD_g,$$

175 (7)

where $\rho_h$, $\rho_g$ are the bulk densities of graupel and hail, respectively. In the MY2 scheme, they are predefined as $\rho_h$= 900 kg m$^{-3}$, and $\rho_g$= 400 kg m$^{-3}$. Integrating Eq. (7) we can get:

$$Q_{fill} = \frac{\pi}{6}(\rho_h - \rho_g) N_{0g} e^{-\lambda_g D^*}(\frac{D^{*3}}{\lambda_g} + \frac{3D^{*2}}{\lambda_g^2} + \frac{6D^*}{\lambda_g^3} + \frac{6}{\lambda_g^4}).$$

(8)

In the modified MY2 scheme, we specify that graupel particles larger than D* undergo wet growth and convert to hail only when

$$(Q_{rg} + Q_{cg})dt \geq Q_{gwet}dt + Q_{fill},$$                                  (9)

where dt is the model integration timestep. D* ranges from 0.5 mm to 25 mm, with a 0.5 mm interval, and the inequality condition (9) is checked for each value. When D* satisfies the condition, but (D* - 0.5 mm) does not, the final D* is

determined through linear interpolation. The mass and concentration of particles larger than D* in the graupel spectrum are then transferred to the hail category. The mass of graupel that converted to hail is calculated as that in the original MY2 scheme, which is:

$$Q_{cngh} = \frac{\pi}{6}\rho_g N_{0g} e^{-\lambda_g D^*}(\frac{D^{*3}}{\lambda_g} + \frac{3D^{*2}}{\lambda_g^2} + \frac{6D^*}{\lambda_g^3} + \frac{6}{\lambda_g^4}).$$                (10)

The modified Milbrandt scheme is referred to as MY2_Mod, while the original one is referred to as MY2_Ori.

**2.3 Cases overview**

We selected three typical heavy rainfall events within Meiyu front system and three hail events that occurred in East China between 2021 and 2024 to evaluate the hail simulation performance of the MY2 scheme. The heavy rainfall events took place on 23–24 June 2022 (Rainstorm1), 17–18 June 2023 (Rainstorm2), and 6–7 July 2023 (Rainstorm3). As shown in Figure 1, the synoptic conditions of Rainstorm1 and Rainstorm3 are quite similar: they are both located in relatively flat

westerly belts, situated south of the 500-hPa trough and the northwest to the western Pacific subtropical high. These two heavy rainfall events are characterized by abundant water vapor, with the precipitable water exceeding 64 and 56 mm, respectively. Combined with low-level southwest moisture transport, the 12-hour cumulative precipitation reached a peak of 189.9 mm in Rainstorm1 (from 1200 on June 23 to 0000 on 24 June 2022, UTC) and 235.1 mm in Rainstorm3 (from 1800 on July 6 to 0600 on 7 July 2023), with the maximum hourly rainfall reaching 118.8 mm h$^{-1}$ and 87.3 mm h$^{-1}$, respectively.





Rainstorm2 also occurred in a relatively flat 500-hPa westerly belt to the northwest of the western Pacific subtropical high, but the synoptic-scale forcing to its north was less pronounced than in the other two cases, and the precipitable water was also slightly lower, mostly ranging between 48–56 mm. The maximum hourly rainfall in this event reached 64.6 mm, with a 12-hour cumulative precipitation of 149.8 mm (from 1800 on 17 July to 0600 on 18 July 2023).

The three selected hail events occurred on 30 April 2021 (Hailstorm1), 24 June 2022 (Hailstorm2), and 25 March 2024

(Hailstorm3). The synoptic conditions of Hailstorm1 and Hailstorm2 are similar, with both located in the northwesterly flow behind the 500-hPa trough and characterized by relatively low precipitable water, which were below 16 mm and 24 mm respectively. In Hailstorm1, a supercell formed in southern Shandong Province at 0700 UTC (1500 LST) on 30 April 2021, and gradually developed into a squall line, moving from northwest to southeast. This hail event brought strong winds (with maximum gusts reaching 45.4 m s$^{-1}$), hail (with a maximum diameter of 3–5 cm), and moderate rainfall (with a maximum

hourly rainfall of 37.5 mm) to Shandong and Jiangsu provinces. In Hailstorm2, several supercells formed in southern Shandong Province at 0600 UTC (1400 LST) on 24 June 2022, and then gradually moved southward, bringing short-term heavy rainfall (with a maximum hourly rainfall of 61.5 mm) and large hailstones with diameters of 5–8 cm. In contrast to these two hail events, which occurred in the northwesterly flow behind the 500-hPa trough, Hailstorm3 developed in the southwesterly flow ahead of a short trough at 500 hPa, with more precipitable water (40–48 mm). In this event, a supercell

formed in western Zhejiang Province at 0600 UTC (1400 LST) and gradually developed into a squall line, affecting the entire province from west to east. This hail event produced short-term heavy rainfall (with a maximum hourly rainfall of 49.9 mm) and large hailstones with maximum diameters of 5–6 cm.

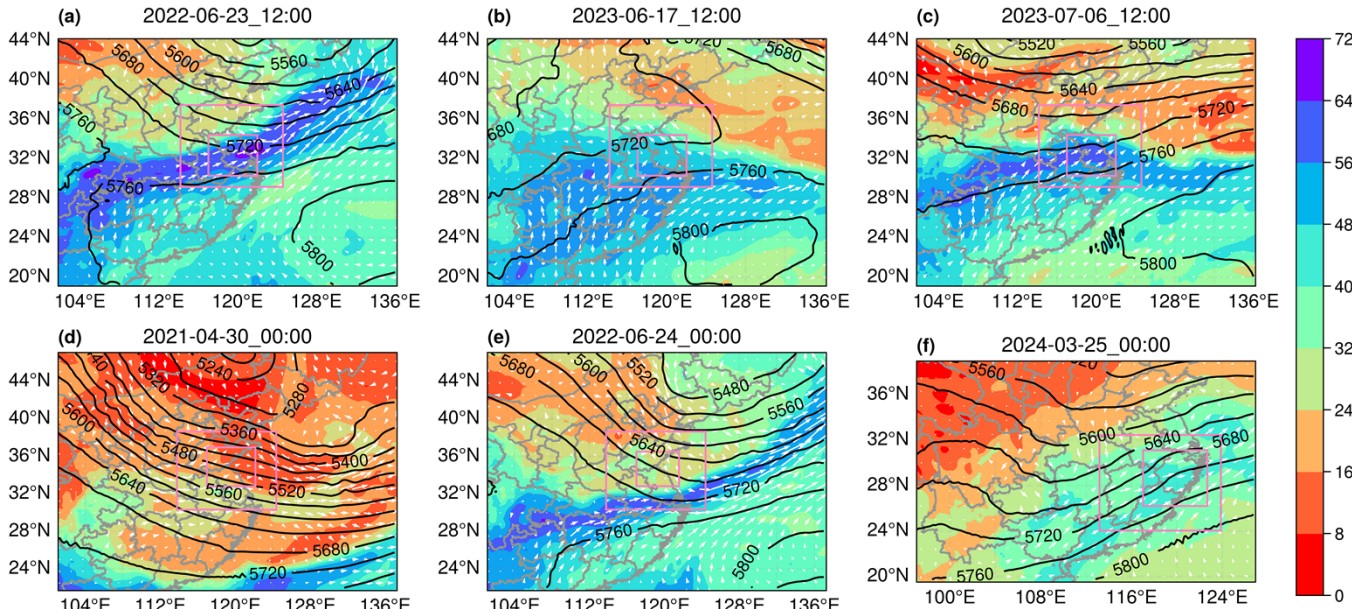

**Figure 1: Synoptic conditions obtained from the ERA5 reanalysis at 1200 UTC on (a) 23 June 2022, (b) 17 June 2023, (c) 6 July 2023, and 0000 UTC on (d) 30 April 2021, (e) 24 June 2022, and (f) 24 March 2024. Colour shadings represent precipitable water**





(unit: mm). Black solid lines represent geopotential height (unit: gpm). White arrows represent wind field at 850 hPa. Pink boxes in each subfigure represent the location of the two innermost domains.

# 3 Results

Figure 2 presents a comparison between observed and simulated column-maximum radar reflectivity for three heavy rainfall
events. The observed radar echoes for these events exhibit an east-west orientation, presenting a typical feature of Meiyu front system. The model successfully reproduces this east–west distribution of radar echoes. In addition, the location of the simulated radar echoes closely aligning with the observations. This indicates that the numerical model successfully captures the main characteristic of these three Meiyu front precipitation events, making the simulation results reliable for further analysis.


Figure 2: Comparison of the observed (upper row) and simulated (bottom row) column-maximum radar reflectivity at (a, d) 1720 UTC on 23 June 2022, (b, e) 2040 UTC on 17 June 2023, and (c, f) 1800 UTC on 6 July 2023. The "×" symbol indicates the presence of at least on point in the vertical direction where the reflectivity factor of hail exceeds the sum of all other hydrometeors. Black lines represent the locations of the vertical cross-sections shown in Fig. 3. Gray lines represent provincial boundaries.



As the observation radar used is a dual-polarization radar, its polarimetric radar variables, such as radar reflectivity, differential reflectivity ($Z_{DR}$), specific differential propagation phase ($K_{DP}$), could be used for hydrometeor classification. The first row in Fig. 2 shows the 2-D classification result, with "×" indicating the presence of identified hailstones in the vertical direction at that position. However, as shown in the figure, no hail was identified in all the three heavy rainfall events.

It should also be noted that the absence of hail, as mentioned above, refers to the non-detection of hail using dual-polarization radar data. This does not imply that there was not even a single hail particle or hail embryo in the real convective clouds. It just indicates that, in these three events, the convective clouds did not contain enough large or numerous hailstones that could be detected by dual-polarization radar. In the MY2 scheme, the hail category represents ice-phase particles with higher density and faster terminal velocities, which are not entirely equivalent to the hail identified

through HCA in observations. To align with the hail identification criteria used in observations, we stipulate that simulated hail in the model is considered to exist at a model grid only when it meets the following three conditions: (1) its mass mixing ratio exceeds 0.1 g kg$^{-1}$, (2) its mass-weighted mean diameter exceeds 2 mm and is larger than the graupel mass-weighted mean diameter at the same model grid, and (3) its equivalent reflectivity factor (Ze) must exceed the combined Ze of all other hydrometeor species (cloud, rain, ice, snow, and graupel). It means that only points where hailstones have a non-

negligible mass, are larger than graupel particles, and contribute the most to the total radar reflectivity are identified as positions where hail is present. This method is in principle similar to, though not identical to, the hail identification algorithm adopted in observations. We did not use the method of first calculating the dual-polarization variables of the simulation results with a radar simulator and then identifying hail using the same hydrometeor classification algorithm as in the observations, since this process would introduce additional errors.

The second row in Fig. 2 illustrates that the simulations using the original MY2 scheme significantly overestimated hail amounts. Hail was present in most areas where the column-maximum reflectivity exceeded 40 dBZ, contrary to observations that detected no hail. Figure 3 illustrates the ratio of hail occurrence in regions where the column-maximum reflectivity exceeds 40 dBZ, comparing observations with simulations across different periods of three heavy rainfall events. In the model, hail was detected in approximately 40% of these regions, with this ratio remaining consistent for 12 hours during all

three events. In contrast, observations showed that this ratio remained near zero for extended periods. The persistent overestimation of hail across multiple cases and over prolonged durations suggests that the MY2 scheme's tendency to overpredict hail is a systematic and recurring issue, rather than an isolated occurrence in specific moments or cases.





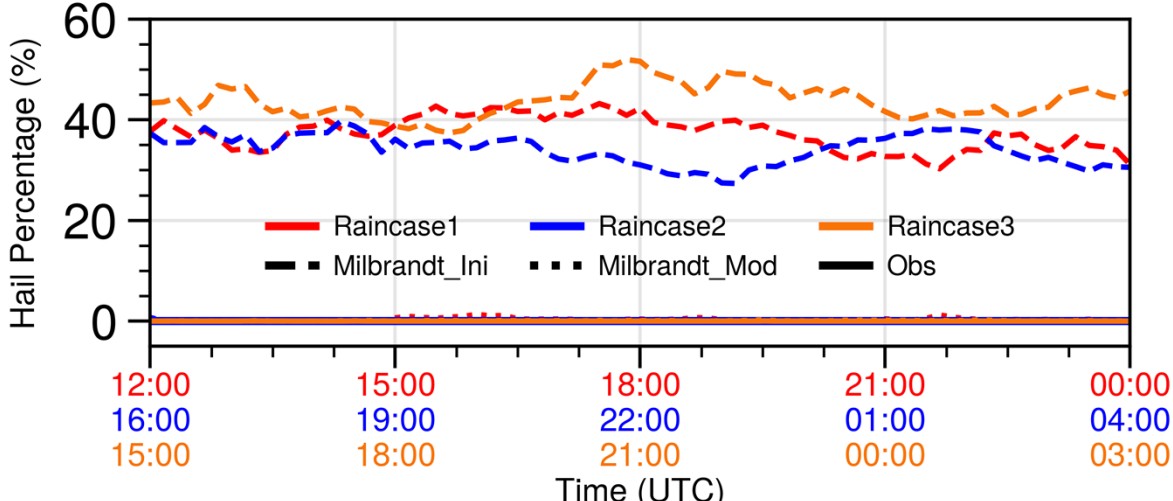

**Figure 3: Comparison of hail percentages in areas where column-maximum radar reflectivity exceeds 40 dBZ between observations (solid lines) and simulations using the original (dashed lines) and modified (dotted lines) Milbrandt 2-moment schemes. Red, blue, and orange lines correspond to Rainstorm 1, 2 and 3, respectively. The red, blue, and orange x-axis labels represent time spans from 1200 UTC on 23 June to 0000 UTC on 24 June, 1600 UTC on 17 June to 0400 UTC on 18 June, and 1500 UTC on 6 July to 0300 UTC on 7 July.**

Figure 4 further compares the observed and simulated radar reflectivity and the spatial distribution of graupel and hail

particles in the vertical direction. Similar to the previous method, regions identified as graupel in the observations are

marked with dots. In the simulations, dots represent regions where the graupel mass mixing ratio exceeds 0.1 g kg⁻¹ and its

equivalent reflectivity factor exceeds the sum of all the other hydrometeors (cloud, rain, ice, snow, and hail). Although the

simulated radar echoes resemble the observed ones structurally, some discrepancies are present. Observed radar reflectivity

values exceeding 40 dBZ are mainly concentrated below 6 km. In contrast, in the simulations, while most regions with

reflectivity over 40 dBZ are also below 6 km, echoes exceeding 40 dBZ are occasionally found above this level. Significant

differences are also noted in the vertical distribution of graupel and hail. In the observations, no hail is detected in the

convection, and graupel particles are primarily located above 5000 m (near the 0 °C level) in regions with reflectivity of 30

dBZ or higher. Even in deep moist convection, where the 30 dBZ echo top height reached nearly 12 km (Fig. 3c), no hail

was detected. In contrast, the simulations show that while graupel is present in the upper parts of the convection, a

substantial amount of hail appears above the 0 °C level, forming several column-shaped regions with reflectivity exceeding

40 dBZ.



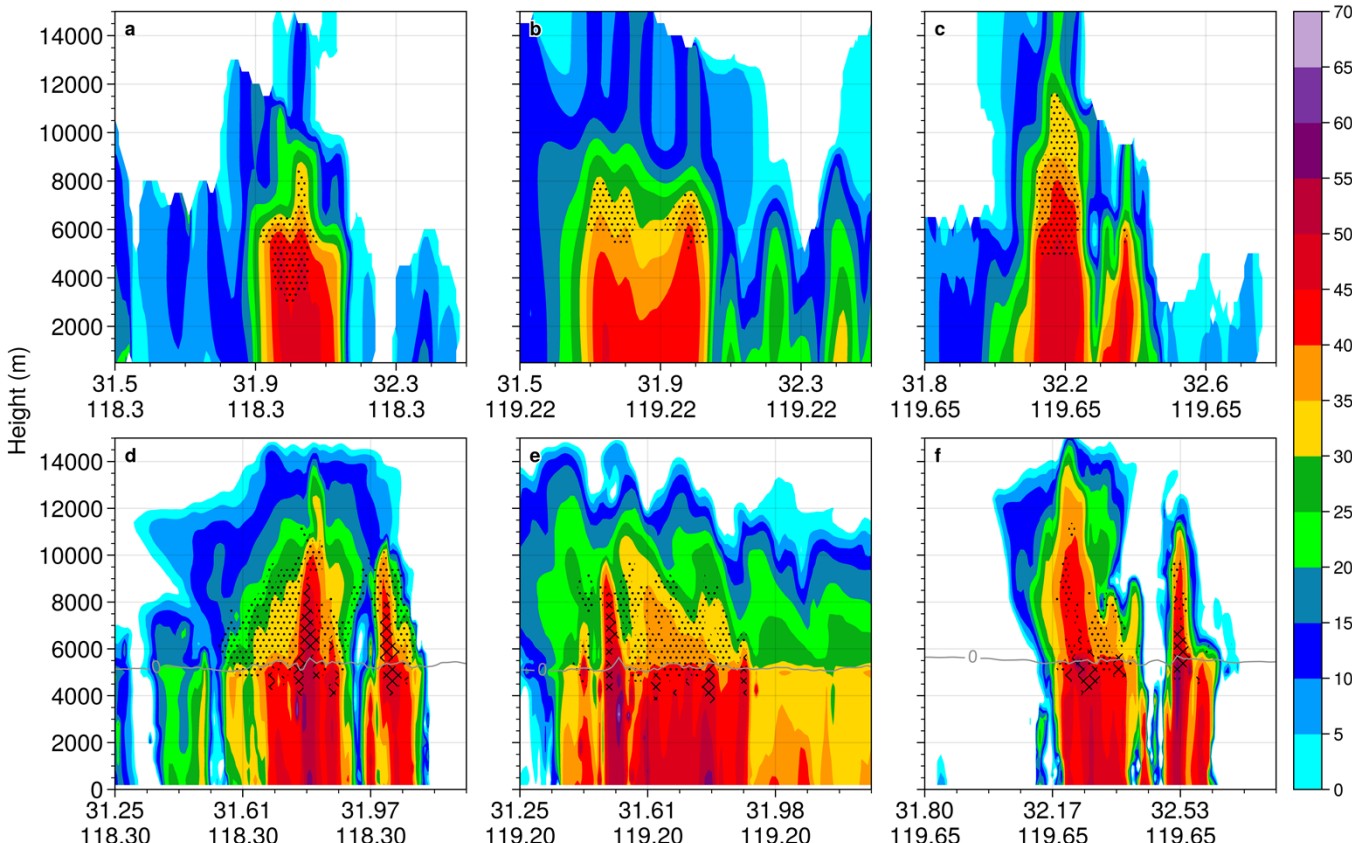

**Figure 4: Comparison of the cross-sections of (upper row) observed and (bottom row) simulated radar reflectivity at (a, d) 1720 UTC on 23 June 2022, (b, e) 2040 UTC on 17 June 2023, and (c, f) 1800 UTC on 6 July 2023. The "×" symbols (dots) indicate that the reflectivity factor of hail (graupel) exceeds the sum of all other hydrometeors at that location. Gray lines represent the 0 °C level.**

Figure 5 further illustrates the reflectivity, mass mixing ratio, mass-weighted mean diameter, and particle size distribution of graupel and hail in three heavy rainfall cases. The reflectivity of both graupel and hail is calculated according to the MY2 scheme, based on the predicted mass and concentration mixing ratios. In these three heavy rainfall cases, graupel predominantly occurs above the 0 °C layer, with mass mixing ratios reaching up to 7 g kg⁻¹. In regions where the its mass mixing ratio exceeds 1 g kg⁻¹, the corresponding reflectivity typically ranges from 25 to 45 dBZ. The mass mixing ratio of hail is smaller than that of graupel, peaking at only 3 g kg⁻¹ in the figure. However, its radar reflectivity often exceeds 40 or 45 dBZ. Hail with a mass mixing ratio greater than 0.1 g kg⁻¹ is only observed above 3000 m, indicating that hail is rare in the lower layers and at the surface. Based on the criteria set in this study, it can be concluded that hail does not reach the surface in the model, despite the fact that the reflectivity of hail consistently extends from aloft to the surface, with values exceeding 40 dBZ. The hail reflectivity below 4 km may arise from the excessive size sorting of hail particles, as suggested by Milbrandt and Yau (2006b) and Milbrandt et al. (2021). Notably, in some regions of the three cases, hail unexpectedly appears below the 0 °C layer, while there is no hail present above the 0 °C layer, where only graupel exists. This suggests




that in the model, hail may form from graupel as it passes through the 0 °C layer, which contradicts the widely accepted hail
formation mechanisms.

The third column presents the mass-weighted mean diameter of graupel ($D_{mg}$) and hail ($D_{mh}$). In regions identified as hail in
Fig. 4, $D_{mh}$ is generally above 3 mm, while graupel has a mass-weighted diameter of around 2 mm. It indicates that
hailstones are larger than graupel particles. The presence of larger hailstones is also evident in the particle size distributions
(PSDs) shown the fourth column. Although the total concentration of hail particles is much lower than that of graupel, the
spectrum width of hail is larger. The concentration of hail particles larger than 5 mm at the three selected points is 4.86, 1.76,
and 1.09 m$^{-3}$, while the concentration of graupel particles larger than 5 mm does not exceed 0.2 m$^{-3}$. Figure 5 indicates that
although hail does not reach the surface in the three heavy rainfall cases in the model, there are indeed sufficiently large
number of hail particles with diameters exceeding those of graupel in the regions aloft that are identified as containing hail,
further supporting the notion that the overforecasting of hail in the MY2 scheme is a real issue.

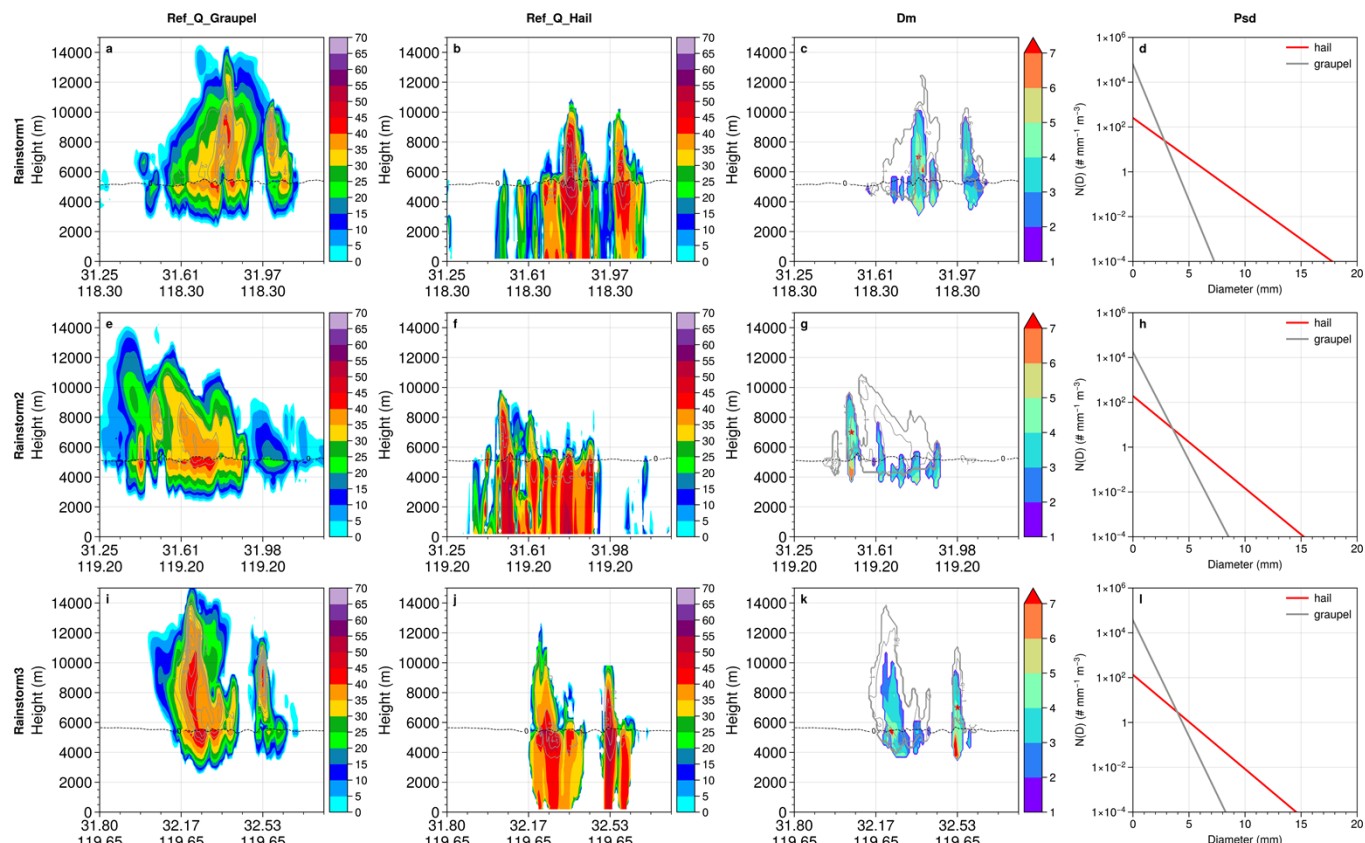


**Figure 5: Recalculated radar reflectivity (color shading, units: dBZ) of graupel (first column) and hail (second column), mass-weighted mean diameter (third column, units: mm) of graupel (gray solid line) and hail (color shading), and particle size distribution (fourth column) of graupel (gray line) and hail (red line) for Rainstorm 1 (top row), 2 (middle row), and 3 (bottom row). Red stars in the third column indicate the locations of the plotted PSDs in the fourth column. Gray dashed lines in the first**
**three columns represent the 0°C layer.**





The overprediction of hail also leads to a misunderstanding of the mechanisms of precipitation formation in the Meiyu system. Figure 6 presents the contribution of various microphysical processes to rain water in regions where rain rate exceeds 20 mm h⁻¹ for the three heavy rainfall cases. The primary source of rainwater in all three cases is the accretion of rainwater by collecting cloud droplets, with median contributions of 51.1% (Rainstorm1), 60.6% (Rainstorm2), and 62.8%

(Rainstorm3), respectively. This is reasonable, as these cases all have a relatively large precipitable water (as shown in Fig. 1), and the warm layer with temperature higher than 0 °C is sufficiently deep (over 5 km). However, in all three cases, the median contribution of hail melting to rainwater is 22.5% (Rainstorm1), 17.7% (Rainstorm2), and 17.5% (Rainstorm3), respectively, which is comparable to the contribution from graupel melting (20.6%, 18.3%, and 17.8%). This suggests that, in the model, hail melting is the second-largest contributor to rainwater. This is in stark contrast to the observations, where

hail was not detected at all.

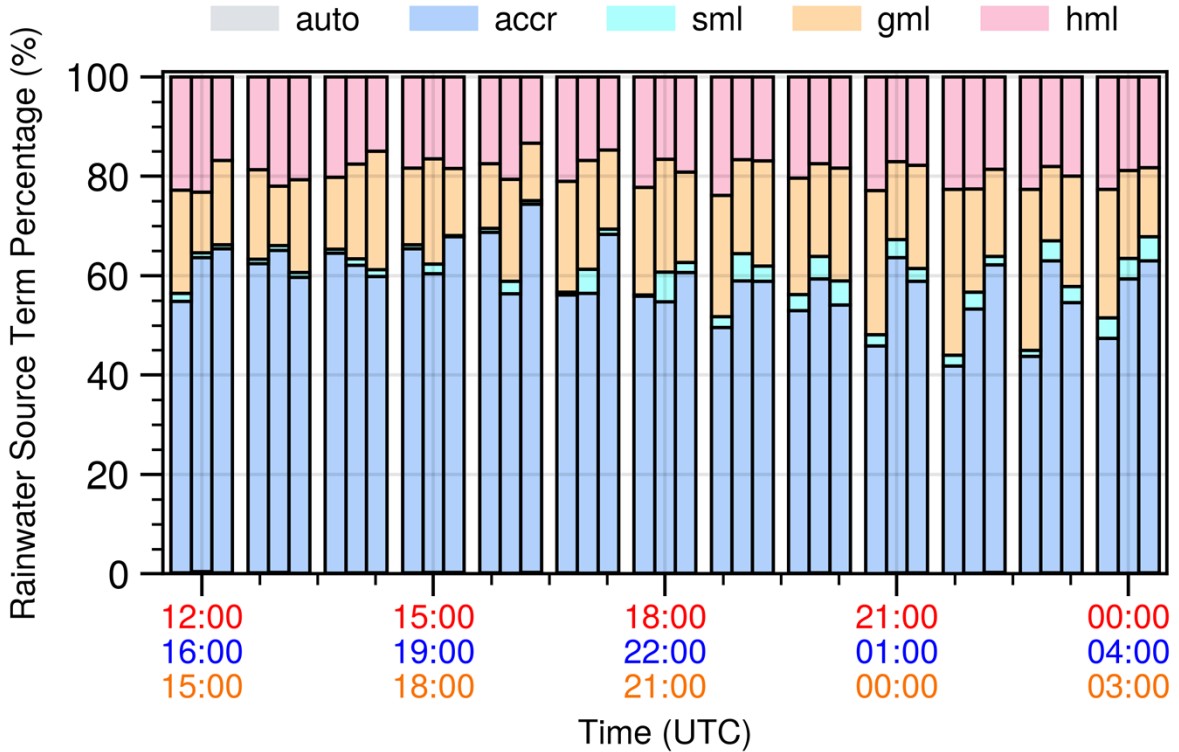

**Figure 6. Contribution of various microphysical processes to rain water in regions where rain rate exceeds 20 mm h⁻¹ for the three heavy rainfall cases. "auto" stands for the auto-conversion of cloud droplet to rainwater, "accr" represents the accretion of rain by collecting cloud droplets, "sml", "gml", "hml" are the melting of snow, graupel and hail to form rain, respectively. The red, blue,**
**and orange x-axis labels represent time spans from 1200 UTC on 23 June to 0000 UTC on 24 June, 1600 UTC on 17 June to 0400 UTC on 18 June, and 1500 UTC on 6 July to 0300 UTC on 7 July.**

Where do these overpredicted hailstones originate? We investigated the hail formation processes in the MY2 scheme, including the graupel-to-hail conversion process, raindrop freezing process, and three-component freezing of raindrops that



lead to hail formation. The result shows that, in terms of mass mixing ratio, the conversion of graupel to hail is the dominant

pathway for hail formation. In the three heavy rainfall cases, this process accounted for 99.8%, 98.1%, and 99.9% of hail

formation, respectively (not shown). In the MY2 scheme, graupel particles are converted to hail once wet growth occurs.

Wet growth happens when graupel particles accrete an excessive amount of supercooled water, releasing a large amount of

latent heat of freezing that prevents all the supercooled water on the surface from freezing. The remaining unfrozen

supercooled water forms a liquid film on the particle surface. For a given particle size, whether wet growth occurs depends

on the environmental temperature, humidity, and supercooled water content. It means that under the same environmental

condition, whether wet growth occurs depends on the particle size, with the critical diameter for wet growth referred to as

Shumann-Ludlam limit (SLL; Young 1993). SLL is the diameter of a particle at which thermal equilibrium is achieved,

assuming the particle surface is maintained at 0 °C. This equilibrium condition accounts for the heat balance of the sensible

heat from the collected supercooled water, the sensible heat exchange between the particle surface and the surrounding air,

and the latent heat of freezing and vapor diffusion at the particle surface. In the MY2 scheme, SLL is approximately

calculated as follows:

$$D_{h0} = 0.02 \times (e^{\frac{-T_c}{1.1 \times 10^4 (q_c + q_r) + 1}} - 1),$$

(11)

where $q_c$ and $q_r$ represent the mass content of supercooled cloud and rain (unit: kg m$^{-3}$), respectively, and $T_c$ denotes

temperature (unit: °C). For a group of graupel particles, particles smaller than SLL continue to undergo dry growth, while

those larger than SLL undergo wet growth and are converted to hail.

The aforementioned approach of determining whether graupel particles convert to hail based on whether wet growth occurs

is physically grounded and has been adopted by many other microphysics scheme development researches, including those

by Ziegler (1985), Mansell et al. (2010), and Tsai and Chen (2020). However, this method has a drawback: although $D_{h0}$ is a

function of both temperature and supercooled water content, it becomes increasingly sensitive to temperature as it

approaches the 0 °C level. As shown in Khain et al. (2010b), $D_{h0}$ remains nearly constant at approximately 0.5 mm at

temperatures around -2 °C, regardless of the liquid water content (Fig. 14 in their study). It means that near -2 °C, any

graupel particle larger than 0.5 mm would undergo wet growth and converts to hail, no matter how much supercooled water

it encounters, which is quite unreasonable.

Figure 7 illustrates the temperature and graupel size when the graupel-to-hail conversion process occurs in three heavy

rainfall cases simulated with the original MY2 scheme. As shown in the figure, approximately 90 % of the graupel-to-hail

conversions occur at temperatures above -0.5 °C in all three cases, and about 45% of the graupel particles are less than 1 mm

in diameter when they are converted to hail. This phenomenon of graupel particles being too easily converted to hail near the

0 °C level is attributed to the imperfect graupel-to-hail conversion parameterization mentioned above.



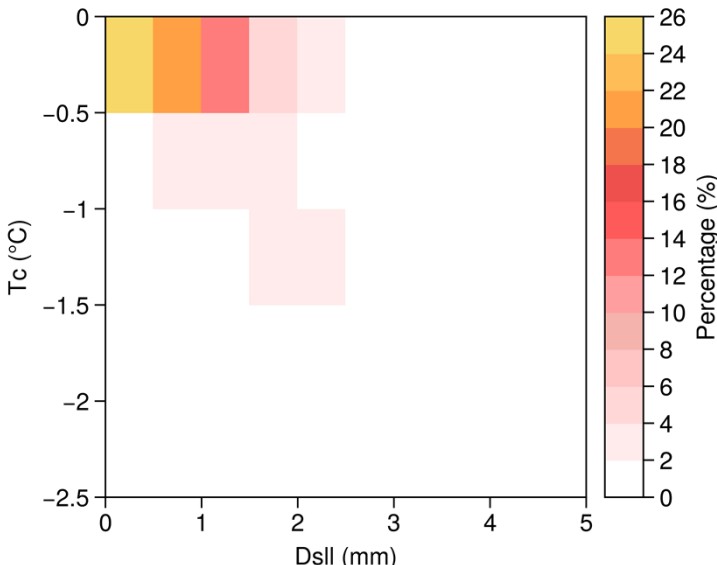

**Figure 7. Two-dimensional frequency distribution of the location where the conversion from graupel to hail occurred as a function of temperature and the derived Schumann-Ludlam Limit.**

Mansell et al. (2010) has identified issues with this graupel-to-hail conversion parameterization method and introduced several artificial constraints to reduce hail overforecasting. For instance, they excluded the process of graupel collecting supercooled raindrops when calculating $D_{h0}$, limited the conversion amount to no more than one-tenth of the graupel mass, and prevented conversions at temperatures warmer than -2 °C. Although these artificial constraints in the NSSL scheme help to reduce hail overestimation to some extent, they inevitably result in underestimation of hail occurrence and smaller hail sizes in the simulations, as indicated in Labriola et al. (Fig. 4f in 2019a, Fig. 2e in 2019b).

Unlike the NSSL scheme, which imposes artificial constraints lacking a physical basis, we introduce the physically grounded spongy wet growth process (described in Section 2.2) to address the overprediction of hail formation in the original graupel-hail conversion parameterization in the MY2 scheme. Figure 8 shows the radar reflectivity and spatial distribution of graupel and hail particles for the three heavy rainfall cases simulated with the modified MY2 scheme. Compared to the results from the original MY2 scheme, the simulations provided by MY2_Mod exhibit a successful mitigation of hail overforecasting in both horizontal and vertical directions. This reduction of hail is accompanied by a noticeable decrease in radar reflectivity intensity. The simulation results provided by MY2_Mod show fewer regions with the column-maximum reflectivity greater than 55 dBZ, aligning more closely with the observations. In the vertical direction, regions with reflectivity larger than 35 dBZ above the 0 °C level are primarily composed of graupel rather than hail, which is also similar to those shown in the observations.





**Figure 8: (top row) Column-maximum radar reflectivity and (bottom row) cross-sections of radar reflectivity simulated with the modified MY2 scheme at (a, d) 1720 UTC on 23 June 2022, (b, e) 2040 UTC on 17 June 2023, and (c, f) 1800 UTC on 6 July 2023. The "×" symbols (dots) indicate the presence of hail (graupel) according to the criteria set in the paper. Gray lines represent the 0 °C level. Black lines in the top row represent the location of the cross-sections in the bottom row.**

The modification of the MY2 scheme to suppress the overprediction of hail also leads to a more realistic representation of the precipitation mechanism in the simulated Meiyu systems. After the modification, the primary source of precipitation remains the accretion of raindrops, accounting for 51.5%, 60.3%, and 63.8% in the three heavy rainfall cases, similar to the original MY2 scheme (Fig. 9). The contribution from hail melting is significantly reduced, dropping from 22.5%, 17.7%, and 17.5% to just 0.3%, 0.2%, and 0.1%, respectively. Meanwhile, the contribution from graupel melting increases substantially, rising from 20.6%, 18.3%, and 17.8% to 46.2%, 36.3%, and 34.1%. Although we cannot directly determine the contribution of each microphysical process to the rainwater content from the observations, the absence of hail and the dominance of graupel in regions with strong reflectivity above the 0°C layer in the observations suggest that the modified scheme provides a more realistic representation of the precipitation processes compared to the original version. The reduction of hail is also shown in Fig. 3, where the hail percentages of all the three heavy rainfall cases produced by the modified MY2 scheme are nearly zero for at least 12 hours, aligning with observations. In fact, compared to the original version, the modified MY2



scheme reduces hail mass by a factor of 100 (not shown). However, it should be noted that the significant reduction in hail shown in Fig. 8 is based on the hail classification criteria outlined in the paper, it does not imply that the hail mass produced by MY2_Mod is nearly zero.

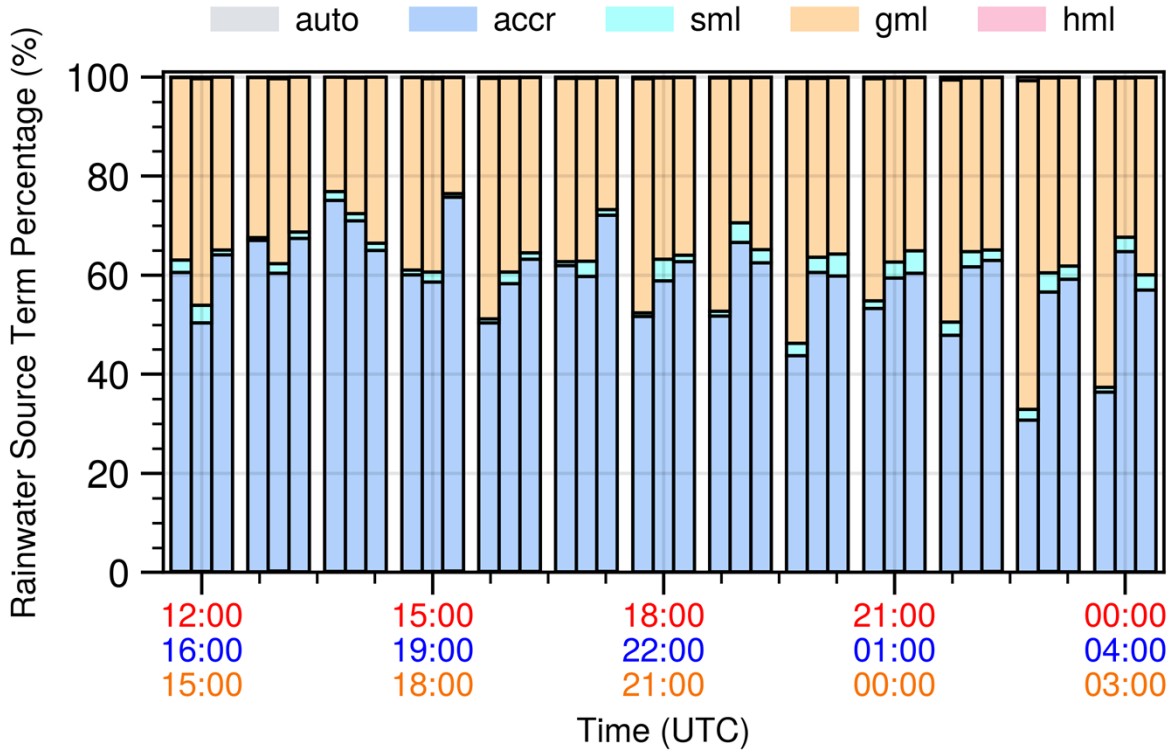


**Figure 9: Same as Fig. 6 but for simulations conducted with the Modified MY2 scheme.**
At this stage, we have developed an improved MY2 scheme that effectively mitigates the overforecasting of hail in heavy rainfall events. However, the question remains whether this scheme can still accurately simulate hail events. While the abovementioned regulation of wet growth in graupel particles is physically justified, it might be overly stringent, potentially

hindering the simulation of hail in real hail events. Thus, we used the MY2_Mod scheme to simulate three hail events that occurred on 30 April 2021, 4 June 2022, and 25 March 2024, respectively. Figure 10 presents the observed and simulated radar reflectivity, along with the spatial distribution of graupel and hail for these events. The results show that the pattern and location of the simulated radar echoes closely match the observations. Moreover, the modified MY2 scheme successfully simulates hail in all three hailstorms, demonstrating that despite the stricter conditions for hail formation, the scheme retains

its ability to produce hail.





**Figure 10: (first two rows)** Observed and **(last two rows)** simulated **(first and third rows)** column-maximum and **(second and fourth rows)** vertical cross sections of radar reflectivity for hail cases on **(first column)** 30 April 2021, **(second column)** 24 June 2022, and **(third column)** 25 March 2024. The "×" symbols (dots) strands for the presence of hail (graupel) according to the criteria set in the paper. Gray lines represent the 0 ℃ level. Black lines in the first and third rows represent the location of the cross-sections in the second and fourth rows, respectively.





The above analysis demonstrates that the modified MY2 scheme could distinguish between heavy rainfall and hail events, a capability absents in the original MY2 scheme and several other microphysics schemes. This modified scheme can therefore be employed to investigate why hail forms in hail events but not in heavy rainfall cases. Figure 11a illustrates the locations

where the graupel-to-hail conversion occurs in three hail cases, alongside a comparison of four associated microphysical processes. The figure shows that graupel-to-hail conversion primarily occurs when, within one integration time step, the mass of supercooled raindrops collected by graupel particles ($Q_{rg}$) exceeds the combined mass of supercooled water required for wet growth ($Q_{gwet}$) and the mass necessary to fill the internal voids of graupel particles to reach hail density ($Q_{fill}$). The conversion of graupel to hail due to sufficient collection with supercooled raindrops, rather than supercooled cloud droplets,

is also justified within the MY2 scheme framework. In the MY2 scheme, the densities of graupel and hail particles are predefined: 400 kg m$^{-3}$ for graupel and 900 kg m$^{-3}$ for hail. To increase the density of graupel to that of hail within a single integration time step, collections with supercooled cloud droplets alone are insufficient. It can only be reached by collecting large supercooled raindrops. Additionally, the amount of supercooled water required to fill the internal voids of graupel particles (approximately 10$^{-6.5}$) is about an order of magnitude greater than that required for graupel wet growth (10$^{-7.5}$). It is

this process that significantly mitigates the overly easy formation of hail in heavy rainfall events.



**Figure 11: (a)** Box plot of four microphysical process conversion rates (unit: $\log_{10}(\text{kg kg}^{-1}\text{ dt}^{-1})$) associated with the conversion process from graupel to hail in three hail events. Dt represents model simulation time step. $Q_{cg}$ and $Q_{rg}$ represent the mass of cloud droplets and raindrops collected by graupel per timestep, respectively. $Q_{gwet}$ stands for the critical mass of supercooled water required for graupel particles to undergo wet growth. $Q_{fill}$ represents the mass of unfrozen water required to fill the internal voids of graupel particles to achieve the bulk density of pure ice. **(b)** Box plot of vertical velocities at locations where the conversion from graupel to hail occurs in (the first three columns) three hail cases, and those at the corresponding locations in (the last three columns) three heavy rain events. **(c)** Comparison of Qr-weighted vertical velocities and terminal fall velocity of rain drops at locations where the conversion from graupel to hail occurs in three hail cases. Qr represents the mass mixing ratio of rain drops. **(d)** The same as (c) but for three heavy rainfall cases at corresponding locations.

In hail events, the vertical velocity at locations where graupel-to-hail conversion occurs is generally greater than 10 m s$^{-1}$.

For example, during the hail event on June 24, 2022, the median vertical velocity reached 16 m s$^{-1}$ (Fig. 11b). Additionally,

the regions where graupel-to-hail conversion occurs are relatively uniformly distributed in areas where radar reflectivity





ranges from 45 to 55 dBZ and temperatures range from -5 °C to 0 °C (not shown). To investigate why graupel-to-hail
conversion does not occur in heavy rainfall cases, we randomly sampled regions in the simulation results of three heavy
rainfall events with radar reflectivity between 45 and 55 dBZ, temperatures between -5 °C and 0 °C, vertical velocity
exceeding 0 m s⁻¹, and the presence of supercooled raindrops. It was found that, in heavy rainfall cases, the vertical velocity
in regions corresponding to those where graupel-to-hail conversion occurs in hail events is significantly lower, generally less
than 10 m s⁻¹. Additionally, in hail events the vertical velocity in graupel-to-hail conversion regions generally exceeds the
terminal fall velocity of raindrops, facilitating the upward transport of raindrops to graupel particles, thereby initiating the
graupel-to-hail conversion process. In contrast, in heavy rainfall events, the vertical velocity does not significantly exceed
the terminal fall velocity of raindrops, limiting the upward transport of raindrops to graupel regions and mitigating the hail
formation process.

Why is the vertical velocity near the 0 °C isotherm in heavy rainfall events insufficient to transport raindrops to higher
levels? Given the relationship between vertical velocity and buoyancy, we analyze the role of buoyancy in both heavy
rainfall and hail events. Buoyancy is calculated as:

$$B = g\left[\frac{\theta - \bar{\theta}}{\theta} + 0.61(q_v - \bar{q_v}) - q_{hydro}\right],$$

(12)

where with $\theta$ being the potential temperature and $q_{hydro}$ the mixing ratios of multiple hydrometeors, respectively. The
overbar in Eq. (12) represents the base state defined as the areal mean in the 100 km × 100 km region enclosed by the gray
dashed lines in insets in the top right corner of each subfigure show in Fig. 9. The explanation for why the gravitational force
of hydrometeors (the last term in Eq. (12)) also affects buoyancy is given below: given the short time required for
hydrometeors to reach their terminal fall velocity during descent, microphysics schemes assume that hydrometeors descend
at a constant terminal velocity, where air resistance is balanced by gravitational force. According to Newton's second law,
the drag force exerted by the hydrometeors on the air (i.e., negative buoyancy) equals the air resistance they encounter,
which is also equal to their gravitational force.

Figure 12 presents vertical profiles of vertical velocity and buoyancy at representative moments and regions for both heavy
rainfall and hail events. The figure demonstrates that the main updraft regions are closely aligned with regions of high
buoyancy resulting from potential temperature differences, indicating that upward motion in both hail and heavy rainfall
events is primarily driven by buoyancy induced by these temperature differences. In hail events, buoyancy within the main
updraft regions typically reaches 0.16 m s⁻², exceeding the average value of about 0.10 m s⁻² in heavy rainfall events.
Consequently, the maximum updraft velocity in hail events (35~40 m s⁻¹) is significantly greater than that in heavy rainfall
events (15~20 m s⁻¹). However, as mentioned before, the vertical velocity near the 0 °C isotherm is more crucial for graupel-
to-hail conversion, thus it is necessary to focus on buoyancy and vertical velocity near the 0 °C level.





**Figure 12: Vertical cross-sections of buoyancy (shaded, unit: m s$^{-2}$) caused by potential temperature differences, vertical velocity (black solid lines), and negative buoyancy due to cloud droplets (violet), raindrops (blue), snow (grape), and graupel (orange) in three (the upper row) heavy rainfall and (the bottom) hail events. The values of negative buoyancy caused by different hydrometeors are contoured at 0.01, 0.02, 0.04, 0.06, and 0.08 m s$^{-2}$. Vertical velocity contours range from 5 to 40 m s$^{-1}$ at intervals of 5 m s$^{-1}$. The gray dashed line indicates the 0 °C isotherm. Insets in the top right corner of each subfigure show the column-maximum radar reflectivity at the corresponding time and the location of the vertical cross-section (black line). The region enclosed by the gray dashed lines represents the area used to calculate the environmental averaged potential temperature.**

In the three heavy rainfall events analysed in this study, the 0 °C isotherm is generally located between 5~5.5 km in altitude. The high mass mixing ratio of raindrops between the 0 °C level and the surface generates negative buoyancy of 0.04~0.08 m s$^{-2}$, which strongly inhibits the updraft below the 0 °C isotherm in heavy rainfall events. In contrast, the situation differs in hail events. For example, in the squall line event on April 30 2021, precipitation occurs not in the main updraft region but rather in the stratiform cloud behind it, preventing the rain water from suppressing the updraft. In the hail cases on June 24 2024 and March 25 2024, while precipitation does occur in the main updraft region, and the negative buoyancy caused by rain drag reaches 0.04 m s$^{-2}$, it does not completely cover the region beneath the main updraft as it does in heavy rainfall events. In hail events, precipitation in the main updraft region typically occurs at the updraft's edges (Fig. 12e), or the updraft itself is slightly tilted (Fig. 12f), preventing the drag force of rainwater from fully weakening the updraft near the 0 °C isotherm.



## 4 Discussion and conclusions

Many microphysics schemes that predefine separate ice-phase hydrometeor categories, such as MY2, NSSL, NTU and SBM

schemes, consider the occurrence of wet growth of graupel particles as a prerequisite for their conversion to hail. While this approach is physically grounded, using the critical amount of supercooled water required to raise graupel surface temperature to 0 °C as the sole trigger for the graupel-to-hail conversion process is insufficient, since less supercooled water is required for the occurrence of wet growth as graupel particles approach the 0 °C level. Consequently, this parameterization method implies that graupel particles are more likely to be converted to hailstones near the 0°C level. In this study, we used the WRF

model configured with the original MY2 scheme to simulate three heavy rainfall cases within the Meiyu front systems in East China, where hail was not detected by hydrometeor classification algorithm based on dual-polarization radar data. It is found that the original MY2 scheme indeed produced a large amount of spurious hail due to the reasons mentioned above. Although these overpredicted hailstones do not reach the surface, the issue of hail overprediction cannot be considered trivial. The overpredicted hailstones, which are larger than the surrounding graupel particles, produce reflectivity values

higher than those observed and contribute excessively to rainfall. This, in turn, complicates our understanding of the radar echo structure, spatial distribution of hydrometeors, and precipitation formation mechanisms in the actual Meiyu system. We argue that the hail overprediction shown in the paper is not confined to the MY2 scheme. Any scheme utilizing the aforementioned graupel-to-hail conversion parameterization method without proper constraints is likely to encounter similar issues of hail overprediction.

By incorporating the spongy wet growth process of graupel particles, we stipulate that graupel-to-hail conversion occurs only when the amount of supercooled cloud water and rainwater collected by graupel particles within one integration time step, exceeds the amount required to maintain their surface temperature at 0 °C, and to fill the internal voids of the graupel particles so that its density reaches the density of pure ice. The modified MY2 scheme greatly mitigated the overprediction of hail. Furthermore, through simulations of three typical hail events in East China, the modified MY2 scheme still

successfully simulated the presence of hail in hail events, demonstrating its ability to distinguish between heavy rainfall and hail processes—an ability that the original MY2 scheme and many other schemes lacked.

We further compared the heavy rainfall events and hail events simulated by the modified MY2 scheme, to analyze why the former does not lead to hail formation, while the latter does. It is found that hail formation primarily occurs when the amount of supercooled rainwater collected by graupel particles exceeds the amount required to fill the internal voids of the graupel

particles. Therefore, the transport of rainwater across the 0 °C level into the subfreezing region is crucial for hail formation. Compared to the three heavy rainfall cases, the three hail cases exhibit stronger updrafts driven by greater positive buoyancy resulting from the temperature difference between the convective updraft and the surrounding environment. Additionally, in the hail cases, the primary updraft region is spatially separated from precipitation, thus the rainwater drag force could not completely inhibit the updraft. In contrast, in the heavy rainfall cases, precipitation directly occurs beneath the convective

updraft. Therefore, the negative buoyancy from rainwater drag weakens the updraft under the 0 °C layer, causing the updraft



velocity to generally be less than the terminal fall velocity of raindrops, preventing them from being transported to subfreezing layer to be collected by graupel particles and forming hail.

The graupel-to-hail conversion parameterization method which incorporates spongy wet growth, is well-suited for microphysics schemes that predefined multiple ice-phase particle categories. This method explains the change in graupel-to-
hail density transition, partially preventing the conversion from graupel to hail from being as abrupt as in the original scheme. Additionally, this parameterization is also applicable to schemes that do not predefine ice-phase particle categories and allow the free evolution of ice-phase particle properties, such as the prognostic-graupel-density MY2 (Milbrandt and Morrison, 2013), different configurations of P3 (Morrison and Milbrandt, 2015; Milbrandt and Morrison, 2016; Milbrandt et al., 2021), and Thompson_Hail (Jensen et al., 2023) schemes. The absence of high-density ice particles above the freezing
level, as shown in the P3 scheme by Labriola et al. (2019b, Fig. 14a in their study) and the Thompson_Hail scheme by Li et al. (2024, Fig. 8f in their study), could potentially be improved by introducing the abovementioned graupel-to-hail conversion parameterization that incorporates the spongy wet growth process.

Previous studies have primarily relied on real hail case studies to evaluate HMSs, focusing on whether these schemes can accurately reproduce radar structures, simulate hail reaching the ground, match observed surface hail locations, and estimate
surface hail size. These studies are crucial as they highlight the sources of simulation biases in hail prediction, and have led to improvements in microphysics schemes. In this paper, we propose an alternative evaluation approach: assessing hail microphysics schemes in heavy rainfall events where no hail is present—not only is there no hail reaching the ground, but there is also no detectable hail aloft, as determined by dual-polarization radar. The goal is to test whether the scheme correctly refrains from simulating hail in such cases. We argue that a robust hail microphysics scheme should accurately
simulate hail in real hail events, while refraining from simulating it in storms where no detectable hail exists. This approach offers valuable insights for the development of microphysics schemes and for advancing our understanding of hail formation. We recommend using these hail-free cases to evaluate other HMSs, including those that predefine ice-phase particle categories (e.g., NSSL, WDM7) and allow particle properties to evolve freely (e.g., different configurations of the P3 scheme). This would help assess their ability to accurately simulate both hail and non-hail processes and identify
potential areas for further model improvements.

*Code and data availability.* The modified MY2 scheme, observation data and code used in this manuscript is available at https://zenodo.org/records/13778097. Code of the standard WRF model (v4.5.1) and the original MY2 scheme can be found at https://github.com/wrf-model/WRF/tree/release-v4.5.1.

*Author contributions.* S.H., G.C. and B.C. designed the research; S.H. performed the research; M.L. and X.X. provided the observation data; S.H. and G.C. analysed the simulation data and wrote the paper.



*Competing interests.* The contact author has declared that none of the authors has any competing interests.


*Acknowledgments.* This work was supported by the National Natural Science Foundation of China (42475009, 42405091), the National Key R&D Program of China (2023YFC3007600) and the CMA Key Innovation Team (CMA2022ZD10).

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
