# Peer review of "Mitigating Hail Overforecasting in the 2-Moment Milbrandt-Yau Microphysics Scheme (v2.25.2\_beta\_04) in WRF (v4.5.1) by Incorporating the Graupel Spongy Wet Growth Process (MY2 GSWG v1.0)"

_EGUsphere, 2024_

## Referee Comment (RC2)

Comments on "Mitigating Hail Overforecasting in the 2-Moment Milbrandt-Yau Microphysics Scheme (v2.25.2_beta_04) in WRF (v4.5.1) by Incorporating the Graupel Spongy Wet Growth Process (MY2_GSWG v1.0)" by Shaofeng Hua, Gang Chen, Baojun Chen, Mingshan Li, and Xin Xu.

Overall, this manuscript is well organized. The authors implemented the spongy wet growth process into a two-moment bulk cloud microphysics scheme in the WRF model to mitigate the overprediction of hail particles. Their approach is based on a thorough consideration of the physical mechanisms of hail formation and successfully reduces hail overprediction in heavy rainfall cases. In addition, the new setting was tested for hail events and was shown to reasonably reproduce hail distributions compared with ground-based radar observations. I have identified several that need clarification, but none of them are serious issues. Some revisions to the organization of the text are also needed to improve readability. Therefore, I recommend minor revision. Please see the detailed comments and suggestions below.

**Main points on organization of text.**

1. To improve readability, I suggest describing the results for the old and new settings in separate subsections.

For example, subsection 3.1 could start at line 224 and subsection 3.2 at line# 374. Some content may need to be slightly revised accordingly.

2. The Discussion and Summary sections should be clearly separated.

The Discussion section may include interpretations, speculations, or opinions, whereas the Summary section should concisely present only what authors did and the key findings derived from the study. After separating these sections, the outcomes will stand out more clearly. In addition, the current Summary section includes too much detailed explanation for a summary. For instance, it may not necessary to rely on the specific case of Maiyu – the overprediction appears to be a more general tendency.

I believe the study's value will still be evident to readers even with a more concise Summary. For example, the last two paragraphs (lines 533-555) are not essential. If the authors consider them necessary, I recommend moving them to the newly separated Discussion section.

**Specific comments.**

1. Please spell out all abbreviations at their first occurrence.
Line#105 MYJ, line#106 ACM2, and line#114 NTU, SBM.

2. Line#140-142. The sentence is overly long, which makes the logical flow unclear. I recommend dividing it into shorter sentences to improve readability.

3. Eqs.(1-10)
Please clarify units and dimensions of Qcg, Qrg, Qgwet, and Qfill. In Eq. (9), Qgwet dt is added to Qfill. This indicates that Qgwet and Qfill have different dimensions, despite both using the same capital letter "Q". This could be quite confusing for readers.

4. Eq.(5)
N(D) is necessary in the integral. In addition, please clarify the assumptions made to derive this formulation (e.g., particle shape and capacitance). Subsequently, check whether these assumptions are consistent with those used for graupel in the MY scheme.
In addition, it should be ≥ instead of =, in my opinion.

5. Line#148. Is Ecg constant?
Please specify its value. In general, Ecg depends on particle sizes and shapes (cf. Böhm, 1999). Some models explicitly consider the dependence using look-up-table or an approximated formulation.

Böhm, J. P., 1999: Revision and clarification of "A general hydrodynamic theory for mixed-phase microphysics." Atmospheric Research, 52, 167–176, https://doi.org/10.1016/S0169-8095(99)00033-2.

6. In Eq. (4), I don't find the terms related to "$|vg(Dg)-vr(Dr)| \approx$ sqrt$\{ (VQg-VQr)^2+0.04V_{Qg}V_{Qr}\}$"

7. Eq. (9)
I think it should be as follows (in fortran coding):
If ( $\{(Qrg+Qcg)dt \geq Qgwetdt\}$ .and. $\{(Qrg+Qcg)dt \geq Q$fill $\}$) then
    Wet growth occurs
Else
    Dry growth occurs
End if

Why do authors add (Qrg+Qcg)dt to Qfill ?

8.  Line#183-185. This is a very good approach.

9.  Figure 1. Please label "Rainstorm1" in the figures as (a) Rainstrom1 (2022-06-23_12:00) to improve readability, as readers may not be particularly interested in the specific date and time of the event here.

10. Line#253-255. The reason provided for not using the radar simulator is not entirely justified. Radar simulators are designed to enable consistent comparisons between observations and models, thereby reducing uncertainties. However, if the authors argue that the numerical model does not incorporate representations of ice particle nonsphericity—an essential factor for sensitivity to polarization—then the benefits of using a simulator would indeed be limited. In that case, citing previous studies (e.g., Matsui et al., 2020) would help clarify and support this point.

Matsui, T., B. Dolan, T. Iguchi, S. A. Rutledge, W. Tao, and S. Lang, 2020: Polarimetric Radar Characteristics of Simulated and Observed Intense Convective Cores for a Midlatitude Continental and Tropical Maritime Environment. J. Hydrometeor., 21, 501–517, https://doi.org/10.1175/JHM-D-19-0185.1

11. Line#293. Please move the location of only as "0.1 g kg-1 is  observed only above 3000 m"

12. Figure 5c, 5g, 5k. It is very difficult to distinguish the gray lines and color shades. Please change the visualization of the figures.

13. Line#354-359. I don't reach the same conclusion from Eq. (11). Please refer to the following figure, which is based on Eq. (11). In addition, this figure does not resemble the one presented in Khain et al. (2010). According to Khain et al., their Figure 14 was derived using a look-up table. Therefore, I conclude that Eq. (11) in this manuscript may not be valid for certain (qc+qr) or Tc ranges. Please clarify specific form of the SLL used in the MY scheme implemented in the WRF model.

[Figure]

14. Line #369-371 (For instance, ~ -2degC). This sentence is difficult to follow due to the dense listing of conditions. Please consider rephrasing it using semi colons and clearer parallel structure to improve readability.

15. Line # 379. "a noticeable decrease" is overemphasized.

16. Line#424. "conditions" is better than "locations".

17. Figure 11b. The boxes are too thin to distinguish colors. Please use wider boxes as in Figure 11a.

18. Line#452-454. This sentence is about heavy rainfall cases. So, the terms "in hail events" is very confusing.

19. Line#466. Fig. 9 is a typo.

20. Line# 473-475. This phenomenon is well described by the cold-pool-shear interaction (e.g., Weisman and Rottuno, 2004).

   Weisman, M. L., and R. Rotunno, 2004: "A Theory for Strong Long-Lived Squall Lines" Revisited. J. Atmos. Sci., 61, 361–382, https://doi.org/10.1175/1520-0469(2004)061<0361:ATFSLS>2.0.CO;2.

---

## Author Comment (AC1)

Reviewer #1

1. Detailed comparisons of the simulations against radar observations have been conducted for hail presence within storms using both the original and modified schemes. However, it is also crucial to examine the corresponding surface hail distribution for both rain and hail events, as accurate surface hail prediction, including hail size and amounts are vital for operational weather forecasting.

Reply: Thanks for your advice. We confirm that no hailstones were observed in the three heavy rainfall cases of this study. For the three hailstorm cases, the maximum hailstone diameters observed on the surface were 3–5 cm (Hailstorm 1), 5–8 cm (Hailstorm 2), and 5–6 cm (Hailstorm 3), respectively. Since our ground hail observations were not obtained from meteorological Bureau but rather from short video platforms like Douyin (TikTok), we only have information on hailstone diameters but lack data on hail amounts. Hail size information is shown in Section 2.3 in the original manuscript.

2. According to dual-polarization radar observations using the HCA, no hail was identified throughout the entire lifespan of the three heavy rainstorms. These findings contradict the simulated rainstorms, which showed a large amount of hail particles even with the modified Milbrandt scheme. Please provide further details on this inconsistency.

Reply: In the model, we defined three criteria for hail identification to ensure that the detected hail signals genuinely represent hail rather than spurious artifacts caused by size sorting: (1) the hail mass mixing ratio must exceed 0.1 g kg⁻¹, (2) its mass-weighted mean diameter must be greater than 2 mm and larger than that of graupel at the same model grid point, and (3) its equivalent reflectivity factor ($Z_e$) must exceed the combined $Z_e$ of all other hydrometeor species (including cloud, rain, ice, snow, and graupel). The absence of detectable hail when applying these criteria with the modified MY2 scheme does not imply a near-zero hail mixing ratio ($q_h$), as the model still contains regions with hail content below 0.1 g kg⁻¹, mass-weighted diameters smaller than 2 mm or reflectivity values lower than those contributed by other hydrometeors. However, these extremely weak hail signals can be considered insignificant for practical purposes. The following is added in the revised manuscript: "*The simulations using the modified MY2 scheme do contain regions with positive hail mass mixing ratios (>0 g kg⁻¹) that fail to meet the three specified hail identification criteria. However, these regions exhibit hail signals below the detection thresholds and are therefore negligible*".

3. The graupel-to-hail conversion in the revised scheme requires a substantial amount of supercooled water collected by graupel at one model time step, specifically at least 1.25 times the mass of graupel particles. This criterion may be too stringent, which explains why it mitigates hail overprediction within rainstorms. However, while hail can be produced within the hailstorms, the simulated radar reflectivity and hail distributions are underestimated compared to radar observations. Additionally, the integration time step also influences the graupel-to-hail conversion, which is also not physically reasonable.

Reply: You are right. Due to the fixed graupel and hail density settings in the MY2 scheme, graupel particles must accrete a substantial amount of supercooled water to fill their internal voids—a quantity exceeding the graupel's own mass. This physical constraint inherently inhibits hail formation and likely explains why the maximum radar reflectivity values in hail cases simulated by the modified MY2 scheme are lower than observed values. However, this discrepancy stems fundamentally from the MY2 scheme's predefined density settings rather than inaccuracies in our implemented spongy wet growth formulation. Were the spongy wet growth process incorporated into a prognostic density scheme like the NSSL scheme (which predicts graupel and hail densities), graupel particles would require significantly less supercooled water accretion for hail conversion. Consequently, the simulated reflectivity patterns would better match the observation.

I concur with your suggestion regarding the influence of the time step ($dt$) on hail conversion. Excessively large dt values may lead to $(Q_{rg}+Q_{cg})*\mathrm{d}t \geq Q_{\mathrm{gwet}}*\mathrm{d}t +Q_{\mathrm{fill}}$, resulting in artificially enhanced hail formation. Therefore, implementing an upper limit for *dt* through comprehensive sensitivity experiments is needed in future studies. However, given our current configuration with 1-km horizontal resolution and 2-s time step, maybe the *dt* is sufficiently small to prevent this overestimation of hail production. The following is added in Discussion: "*Our proposed method of introducing spongy wet growth to improve the graupel-to-hail conversion process still requires further investigation in several aspects. For instance, the model integration timestep may exert an influence on the conversion occurrence. Because larger dt values increase the likelihood that the supercooled water content accreted by graupel particles exceeds the amount required to fill their internal voids, thereby artificially enhancing hail formation. Although our current experiments employ a sufficiently small dt (2 s), comprehensive sensitivity tests are needed in the future to provide optimal dt thresholds. Furthermore, the predefined graupel and hail density in the MY2 scheme physically constrain hail conversion by requiring graupel particles to accrete sufficient supercooled water within a single timestep. This inherent configuration may systematically suppress hail production. Future implementation of this approach within microphysical schemes which could predict the density of graupel and hail particles, such as the prognostic-graupel-density MY2 (Milbrandt and Morrison, 2013) and the NSSL scheme, could potentially mitigate this limitation. Additionally, the absence of high-density ice particles above the freezing level, as shown in the P3 scheme (Morrison and Milbrandt, 2015; Milbrandt and Morrison, 2016; Milbrandt et al., 2021) by Labriola et al. (2019b, Fig. 14a in their study) and the Thompson_Hail scheme (Jensen et al., 2023) by Li et al. (2024, Fig. 8f in their study), could potentially be improved by introducing the spongy wet growth process to increase the density of riming particles*".

4. The discussions in the last two paragraphs of Section 3 lack clarity. For instance, while both thermal buoyancy and dynamic vertical pressure gradient force are important for the vertical motion, only the buoyancy term is investigated.

Reply: Thanks for your comments. We have calculated the dynamic vertical pressure gradient $-\frac{1}{\bar{\rho}}\frac{\partial p'_d}{\partial z}$, where $p'_d$ is dynamic pressure and is calculated as:

$$\nabla^2 p'_d = \bar{\rho}(\mid \boldsymbol{\omega} \mid^2 - \mid D \mid^2)$$

$$\mid \boldsymbol{\omega} \mid^2 = (w_y - v_z)^2 + (u_z - w_x)^2 + (v_x - u_y)^2$$

$$\mid D \mid^2 = (u_x^2 + u_y^2 + u_z^2) + (v_x^2 + v_y^2 + v_z^2) + (w_x^2 + w_y^2 + w_z^2) - (\mathrm{d}\ln \bar{\rho}/\mathrm{d}z)\boldsymbol{v} \cdot \nabla w - (\mathrm{d}^2\ln \bar{\rho}/\mathrm{d}z^2)w^2$$

where $\boldsymbol{\omega}$ is the three-dimensional vorticity vector, $|D|$ is the magnitude of the total deformation, $\bar{\rho}$ is the basic state density. Its magnitude ranges between approximately -0.03 to 0.03 m s⁻², which is an order of magnitude smaller than the buoyancy force (-0.2 to 0.2 m s⁻²). This indicates that the dynamic pressure has a negligible contribution to vertical acceleration compared to buoyancy. Furthermore, the strong spatial correlation between updrafts and positive buoyancy regions (Figure 12) further confirms that vertical motion is primarily driven by buoyancy. The following is added in the revised manuscript: "*Given that the dynamic vertical pressure gradient force (about 0.03 m s⁻²) is about an order of magnitude smaller than the buoyancy force (about 0.2 m s⁻²) in these six cases (not shown), our study primarily focuses on the dynamical effects of buoyancy*".

[Figure]

Figure R1. Similar to Figure 12 but for dynamic vertical pressure gradient force (color shading, unit: m s⁻²). Vertical velocity (black lines) contours range from 5 to 40 m s⁻¹ at intervals of 5 m s⁻¹.

---

## Author Comment (AC2)

Reviewer #2

Comments on "Mitigating Hail Overforecasting in the 2-Moment Milbrandt-Yau MicrophysicsScheme (v2.25.2_beta_04) in WRF (v4.5.1) by Incorporating the Graupel Spongy Wet Growth Process (MY2_GSWG v1.0)" by Shaofeng Hua, Gang Chen, Baojun Chen, Mingshan Li, and Xin Xu.

Overall, this manuscript is well organized. The authors implemented the spongy wet growth process into a two-moment bulk cloud microphysics scheme in the WRF model to mitigate the overprediction of hail particles. Their approach is based on a thorough consideration of the physical mechanisms of hail formation and successfully reduces hail overprediction in heavy rainfall cases. In addition, the new setting was tested for hail events and was shown to reasonably reproduce hail distributions compared with ground-based radar observations. I have identified several that need clarification, but none of them are serious issues. Some revisions to the organization of the text are also needed to improve readability. Therefore, I recommend minor revision. Please see the detailed comments and suggestions below.

Reply: We sincerely appreciate your thorough and constructive comments on our manuscript "*Mitigating Hail Overforecasting in the 2-Moment Milbrandt-Yau Microphysics Scheme (v2.25.2_beta_04) in WRF (v4.5.1) by Incorporating the Graupel Spongy Wet Growth Process (MY2_GSWG v1.0)*". We have carefully considered all your valuable suggestions and provided a point-by-point response to each comment. Your insightful suggestions have significantly improved the clarity, organization, and scientific rigor of our work. Your thoughtful feedback has not only strengthened our manuscript but also deepened our understanding of the subject. We believe the revised version addresses all your concerns and hope it meets the journal's standards. Thank you again for your time and valuable input.

Main points on organization of text.

1. To improve readability, I suggest describing the results for the old and new settings in separate subsections. For example, subsection 3.1 could start at line 224 and subsection 3.2 at line# 374. Some content may need to be slightly revised accordingly.

Reply: Thanks for your advice. Section 3 (Results) has been divided into three subsections—"3.1 Results with the original MY2 scheme", "3.2 Results with the modified MY2 scheme", and "3.3 Hail formation mechanism"—to improve readability.

2. The Discussion and Summary sections should be clearly separated.

   The Discussion section may include interpretations, speculations, or opinions, whereas the Summary section should concisely present only what authors did and the key findings derived from the study. After separating these sections, the outcomes will stand out more clearly. In addition, the current Summary section includes too much detailed explanation

for a summary. For instance, it may not necessary to rely on the specific case of Maiyu – the overprediction appears to be a more general tendency.

I believe the study's value will still be evident to readers even with a more concise Summary. For example, the last two paragraphs (lines 533-555) are not essential. If the authors consider them necessary, I recommend moving them to the newly separated Discussion section.

Reply: Thanks for your advice. We have restructured the final section into two distinct parts: Conclusion and Discussion, to ensure the Conclusion remains focused on summarizing our core findings. The original last two paragraphs have been slightly revised and moved to the Discussion section, where we highlight potential directions for future improvements to this study.

Specific comments.

1. Please spell out all abbreviations at their first occurrence. Line#105 MYJ, line#106 ACM2, and line#114 NTU, SBM.

Reply: Thanks for your advices. We have included the full terms of all abbreviations in the revised manuscript, 'MYJ', 'ACM2', 'NSSL', 'NTU' and 'SBM' refer to 'Mellor-Yamada-Janjic', 'Asymmetric Convective Model version 2', 'National Severe Storms Laboratory', 'National Taiwan University', and 'Spectral Bin Microphysics', respectively.

2. Line#140-142. The sentence is overly long, which makes the logical flow unclear. I recommend dividing it into shorter sentences to improve readability.

Reply: Thanks. It is revised as "*In the modified MY2 scheme, graupel particles are allowed to convert to hail only under specific conditions. This conversion occurs when the total amount of supercooled cloud droplets and raindrops collected by particles larger than a critical size, exceeds the amount required for their wet growth and for filling their internal voids to reach a density of 900 kg m$^{-3}$ within one integration timestep*".

3. Eqs.(1-10) Please clarify units and dimensions of Qcg, Qrg, Qgwet, and Qfill. In Eq. (9), Qgwet dt is added to Qfill. This indicates that Qgwet and Qfill have different dimensions, despite both using the same capital letter "Q". This could be quite confusing for readers.

Reply: Thanks. The units of Qcg, Qrg, and Qgwet are all kg m$^{-3}$ s$^{-1}$, while Qfill is expressed in kg m$^{-3}$. These unit definitions have now been explicitly stated in the revised manuscript.

4. Eq. (5) N(D) is necessary in the integral. In addition, please clarify the assumptions made to derive this formulation (e.g., particle shape and capacitance). Subsequently,

check whether these assumptions are consistent with those used for graupel in the MY scheme. In addition, it should be $\geq$ instead of $=$, in my opinion.

Reply: We appreciate your comment. In response, we have now included the particle size distribution *N(D)* in Eq. (5). And, as noted in Musil (1970) and Milbrandt and Yau (2005a), graupel particles were treated as spherical, hence their capacitance (*C*) equals the particle radius (*r*). We have verified that all configurations and required parameters in Eqs. (1–10) are fully consistent with the original MY scheme. Regarding Eq. (5), which calculates the critical content for graupel wet growth, we retain the equal sign ("=") in the equation for physical correctness. To improve clarity, the following sentences have been incorporated into the revised manuscript: "*In the derivations of Eq. (5) by Musil (1970) and Milbrandt and Yau (2005a), graupel particles were assumed to be spherical, and their capacitance was set equal to the graupel particle radius. Eq. (5) indicates that when the supercooled water content exceeds $Q_{gwet}$, all graupel particles with diameters larger than D\* will undergo wet growth*".

5. Line#148. Is Ecg constant? Please specify its value. In general, Ecg depends on particle sizes and shapes (cf. Böhm, 1999). Some models explicitly consider the dependence using look-up-table or an approximated formulation.

Böhm, J. P., 1999: Revision and clarification of "A general hydrodynamic theory for mixed-phase microphysics."Atmospheric Research, 52, 167–176, https://doi.org/10.1016/S0169-8095(99)00033-2.

Reply: Thanks for your comments. The value of Ecg is consistent with the Milbrandt scheme and is calculated based on the study by Cober and List (1993), which is a function of the mass-weighted mean diameters of cloud droplets and graupel particles. The relevant sentence is reworded as "$E_{cg}$ *is the collecting efficiency between graupel and cloud, which is function of the mass-weighted mean diameters of cloud droplets and graupel particles (Cober and List, 1993)*".

Reference

Cober, S. G., and R. List: Measurements of the Heat and Mass Transfer Parameters Characterizing Conical Graupel Growth. *J. Atmos. Sci.*, 50, 1591–1609, https://doi.org/10.1175/1520-0469(1993)050<1591:MOTHAM>2.0.CO;2, 1993.

6. In Eq. (4), I don't find the terms related to "$|wg(Dg)$—$wr(Dr)|\approx$ sqrt$\{ (VQg$—$VQr)2+0.04VQgVQr\}$"

Reply: We apologize that there was an error in Eq. (4). We have now corrected and rewritten the equation as follows:

$$Q_{rg} = \sqrt{(V_{Qg} - V_{Qr})^2 + 0.04 V_{Qg} V_{Qr}} \, \rho_w \frac{\pi^2}{24} E_{rg} N_{Tr} N_{0g} e^{-\lambda_g D^*} \left[ \frac{\Gamma(4)}{\lambda_r^3} \left( \frac{D^{*2}}{\lambda_g} + \frac{2D^*}{\lambda_g^2} + \frac{2}{\lambda_g^3} \right) \right.$$

$$\left. + \frac{2\Gamma(5)}{\lambda_r^4} \left( \frac{D^*}{\lambda_g} + \frac{1}{\lambda_g^2} \right) + \frac{\Gamma(6)}{\lambda_r^5} \frac{1}{\lambda_g} \right]$$

7. Eq. (9) I think it should be as follows (in fortran coding):

If ( {$(Qrg+Qcg)$dt≥Qgwetd$t$ } .and. {$(Qrg+Qcg)$dt≥$Q$fill }) then

   Wet growth occurs

Else

   Dry growth occurs

End if

Why do authors add (Qrg+Qcg)dt to Qfill ?

Reply: In our code, the approach is as follows:

If (($Q_{rg}+Q_{cg}$)*d$t$ ≥ ($Q_{gwet}$*d$t$ +Q$_{fill}$)) then

   Wet growth occurs

Else

   Dry growth occurs

End if

That means the wet growth of graupel occurs only when the amount of supercooled liquid water it collected (($Q_{rg}+Q_{cg}$)*d$t$) exceeds the sum of the amount required to raise their surface temperature to 0°C ($Q_{gwet}$*d$t$), and the amount needed to fill their internal voids ($Q_{fill}$). We did not add (Q$_{rg}$+Q$_{cg}$)*dt to Q$_{fill}$. For clarity, the following is added in the revised manuscript "*In the modified MY2 scheme, we specify that graupel particles larger than D\* undergo wet growth and convert to hail only when the amount of supercooled*

*liquid water they collected (($Q_{rg} + Q_{cg}$)dt) exceeds the sum of the amount required to*

*raise their surface temperature to 0°C ($Q_{gwet}$dt), and the amount needed to fill their internal voids ($Q_{fill}$)*".

8. Line#183-185. This is a very good approach.

Reply: We appreciate your affirmation.

9. Figure 1. Please label "Rainstorm1" in the figures as (a) Rainstrom1 (2022-06-23_12:00) to improve readability, as readers may not be particularly interested in the specific date and time of the event here.

Reply: Thanks. Figure 1 is replotted according to your suggestion, which is shown below.

[Figure]

Figure R1. The same as Figure 1 in the revised manuscript.

10. Line#253-255. The reason provided for not using the radar simulator is not entirely justified. Radar simulators are designed to enable consistent comparisons between observations and models, thereby reducing uncertainties. However, if the authors argue that the numerical model does not incorporate representations of ice particle nonsphericity—an essential factor for sensitivity to polarization—then the benefits of using a simulator would indeed be limited. In that case, citing previous studies (e.g., Matsui et al., 2020) would help clarify and support this point.

Matsui, T., B. Dolan, T. Iguchi, S. A. Rutledge, W. Tao, and S. Lang, 2020: Polarimetric Radar Characteristics of Simulated and Observed Intense Convective Cores for a Midlatitude Continental and Tropical Maritime Environment. J. Hydrometeor., 21, 501–517, https://doi.org/10.1175/JHM-D-19-0185.1

Reply: Thanks for your advice. The relevant sentence is reworded as "*We did not use the method of first calculating the dual-polarization variables of the simulation results with a radar simulator and then identifying hail using the same hydrometeor classification algorithm as in the observations,* **since MY2 scheme lacks proper treatment of ice particle shapes, which are essential for polarization sensitivity, resulting in limited benefits from using a simulator (Matsui et al., 2020)**".

11. Line#293. Please move the location of only as "0.1 g kg-1 is only observed only above 3000 m"

Reply: Thanks. It is revised.

12. Figure 5c, 5g, 5k. It is very difficult to distinguish the gray lines and color shades. Please change the visualization of the figures.

Reply: We adjusted the y-axis range in the third column of Figure 5 to 3000-12000 m to better display the mass-weighted diameters of graupel and hail particles. Additionally, we plotted the mass-weighted diameters of graupel at 1mm intervals from 2 to10 mm (with fewer lines since graupel diameters rarely exceed 3 mm). The modified figure presented below should more effectively demonstrate that the mass-weighted diameters of hail particles are typically larger than those of graupel particles.

[Figure]

Figure R2. The same as Figure 4 in the revised manuscript.

13. Line#354-359. I don't reach the same conclusion from Eq. (11). Please refer to the following figure, which is based on Eq. (11). In addition, this figure does not resemble the one presented in Khain et al. (2010). According to Khain et al., their Figure 14 was derived using a look-up table. Therefore, I conclude that Eq. (11) in this manuscript may not be valid for certain (qc+qr) or Tc ranges. Please clarify specific form of the SLL used in the MY scheme implemented in the WRF model.

[Figure]

Reply: We sincerely apologize if the presentation of Eq. (11) was unclear, which may have led to a misunderstanding of the formula. Here, *-Tc* should be part of the exponent of the natural exponential *e*, rather than being a numerator divided by *e*'s exponential term. In the revised manuscript, we have added parentheses to the exponent of *e* in Eq. (11) to prevent similar confusion. The revised Eq. (11) is

$$D_{h0}=0.02\times(e^{\left(\frac{-T_c}{1.1\times10^4(q_c+q_r)+1}\right)} - 1)$$

14. Line #369-371 (For instance, ~ -2degC). This sentence is difficult to follow due to the dense listing of conditions. Please consider rephrasing it using semi colons and clearer parallel structure to improve readability.

Reply: Thanks. It is revised as "*For instance, they excluded the process of graupel collecting supercooled raindrops when calculating $D_{h0}$; limited the conversion amount to no more than one-tenth of the graupel mass; and prevented conversions at temperatures warmer than -2 °C.*"

15. Line # 379. "a noticeable decrease" is overemphasized.

Reply: Thanks, "noticeable" is removed.

16. Line#424. "conditions" is better than "locations".

Reply: Thanks, it is replaced.

17. Figure 11b. The boxes are too thin to distinguish colors. Please use wider boxes as in Figure 11a.

Reply: It is replotted as you suggested.

[Figure]

Figure R3. The same as Figure 11 in the revised manuscript.

18. Line#452-454. This sentence is about heavy rainfall cases. So, the terms "in hail events" is very confusing.

Reply: We have adjusted the sentence structure, and the revised version is as follows: "*It was found that, in heavy rainfall cases, the vertical velocity in regions corresponding to those where graupel-to-hail conversion occurs in hail events is significantly lower, generally less than 10 m s⁻¹. Additionally, in heavy rainfall events, the vertical velocity does not significantly exceed the terminal fall velocity of raindrops, limiting the upward transport of raindrops to graupel regions and mitigating the hail formation process. In contrast, in hail events the vertical velocity in graupel-to-hail conversion regions generally exceeds the terminal fall velocity of raindrops, facilitating the upward transport of raindrops to graupel particles, thereby initiating the graupel-to-hail conversion process*".

19. Line#466. Fig. 9 is a typo.

Reply: Thanks, it is revised to "Fig. 12".

20. Line# 473-475. This phenomenon is well described by the cold-pool-shear interaction (e.g., Weisman and Rottuno, 2004).

Weisman, M. L., and R. Rotunno, 2004: "A Theory for Strong Long-Lived Squall Lines" Revisited. J. Atmos. Sci., 61, 361–382, https://doi.org/10.1175/1520-0469(2004)061<0361:ATFSLS>2.0.CO;2.

Reply: The cold-pool-shear interaction is more pronounced in hail cases, particularly in Hailcase1, where a distinct negative buoyancy zone associated with cold pool is observed below 2 km. Although the updraft exhibits slight tilting, it remains predominantly upright, indicating a near-balance state between the cold pool and environmental shear, which is a key factor contributing to this severe hailstorm event. In contrast, cold pools may not be as evident in heavy rainfall cases. In Meiyu frontal systems, abundant moisture likely suppresses rainfall evaporation, thereby weakening cold pool intensity. Consequently, the convective systems producing heavy rainfall under Meiyu fronts may not be primarily driven by cold pool.
The following is added in the revised manuscript: "*In the hailstorm cases, particularly in Hailcase1, a negative buoyancy zone formed by the cold pool is present below 2 km altitude. The quasi-upright updraft structure indicates a dynamic equilibrium between the cold pool and environmental shear, consistent with the RKW theory (Weisman and Rotunno 2004), which favored the sustained severe convection*".